# ERCC6L2 ensures repair fidelity for staggered-end DNA double-strand breaks

Eric J. Aird [1,9], Almudena Serrano-Benitez [2,3,9], Sebastian M. Siegner [1,9], Elda Cannavo[4], Rimma Belotserkovskaya[2,3], Nadia Gueorguieva[2,3], John Fielden[1], Grégoire Cullot[1], Sandra Ammann [5,6], Aldo S. Bader[2,3], Vipul Gupta[3], Geoffroy Andrieux [7], Rebecca Raab [5,6], Mónica Del Rey González[8], Toni Cathomen [5,6], Petr Cejka [4], Jacob E. Corn [1]✉ & Stephen P. Jackson [2,3]✉

DNA double-strand breaks (DSBs) both pose threats to genome integrity and are commonly used for genome editing applications. Structural features of DSB ends play key roles in determining DNA repair pathway usage and outcomes during genome editing, but the cellular factors involved in these processes are only partially known. Through genome-wide CRISPRi screening, we identify ERCC6L2 as critical for repairing Cas12a-induced staggered DSBs but irrelevant for Cas9-induced blunt DSBs. We show that ERCC6L2 acts as a protection factor for staggered DSBs with either 5′ or 3′ polarity, preventing large deletions and translocations stemming from DNA damage induced by Cas12a, TALENs, or dual Cas9 nicks. Furthermore, ERCC6L2 loss hypersensitizes cells to multiple staggered DSBs induced by promiscuous Cas12a activity or etoposide-induced TOP2 trapping. By combining genetics and biochemical reconstitution, we find that ERCC6L2 counteracts MRE11-RAD50-NBS1 (MRN)-mediated resection by binding and melting staggered DNA ends, thereby promoting accurate end joining. Our data reveal a protective role of ERCC6L2 in staggered-end DSB repair, which suggests the molecular underpinnings of pathology in patients with *ERCC6L2* mutations and cautions against using overhang-inducing genome editing tools for their treatment.

The DNA damage response (DDR) is an intricate network that has evolved to coordinate the repair of myriad DNA lesions throughout the lifetime of a human cell. Double-strand breaks (DSBs) are among the most toxic DNA lesions, since they can lead to acentric chromosomes and large-scale loss of genetic information during cell division[1]. DSBs can arise endogenously when a single-strand nick is converted into a DSB during replication[2] or by the abortive activity of DNA topoisomerase II (TOP2) during the release of torsional stress from DNA[3]. Programmed DSBs are also induced during normal biology, for example, in meiotic crossing over, in V(D)J recombination during B- and T-lymphocyte development, and in class-switch recombination (CSR) that mediates antibody isotype switching in activated B cells.

[1]Department of Biology, Institute of Molecular Health Sciences, ETH Zurich, Zurich, Switzerland. [2]Cancer Research UK Cambridge Institute, University of Cambridge, Cambridge, UK. [3]The Gurdon Institute, University of Cambridge, Cambridge, UK. [4]Institute for Research in Biomedicine, Università della Svizzera italiana (USI), Faculty of Biomedical Sciences, Bellinzona, Switzerland. [5]Institute for Transfusion Medicine and Gene Therapy, Medical Center – University of Freiburg, Freiburg, Germany. [6]Center for Chronic Immunodeficiency, Faculty of Medicine, University of Freiburg, Freiburg, Germany. [7]Institute of Medical Bioinformatics and Systems Medicine, Medical Center - University of Freiburg, Faculty of Medicine, University of Freiburg, Freiburg, Germany. [8]Instituto de Investigación Biomédica de Salamanca (IBSAL), Centro de Investigación del Cancer (CIC), Salamanca, Spain. [9]These authors contributed equally: Eric J. Aird, Almudena Serrano-Benitez, Sebastian M. Siegner. ✉e-mail: jacob.corn@biol.ethz.ch; Steve.Jackson@cruk.cam.ac.uk

DSBs are also purposefully introduced and harnessed to great effect by researchers and clinicians using genome editing tools such as Cas9.

Eukaryotic cells have developed sophisticated mechanisms to resolve DSBs accurately and efficiently. These repair pathways both protect the genome and underpin genome editing technologies. DSBs can be repaired through two distinct mechanisms: templated repair copying from a homologous donor DNA molecule (homologous recombination; HR), and more direct DNA-end ligation processes. The latter can further be differentiated into non-homologous end-joining (NHEJ) and microhomology-mediated end-joining (MMEJ)[4]. Efficient and stable synapsis of non-homologous or micro-homologous DNA ends is crucial to facilitate the correct pairing of DNA fragments, as improper alignment of DNA ends can result in inter-chromosomal deletions and chromosomal translocations[5,6].

During NHEJ, different DNA end configurations and their processing play crucial roles in determining the precise repair steps used[7]. For instance, DNA-PKcs (*PRKDC*) regulates the recruitment of the nuclease Artemis (*DCLRE1C*) to facilitate repair of complex DNA ends, such as hairpins formed during V(D)J recombination[8]. Mutations in either Artemis or DNA-PKcs can cause Severe Combined Immunodeficiency (SCID)[9–11], highlighting the importance of proper processing of these end structures. DSBs introduced by genome editing tools can also have various end architectures. For example, Type II Cas enzymes (e.g., *S. pyogenes* Cas9) predominantly produce blunt DSBs while Type V Cas enzymes (e.g., Cas12a and the recently described TnpB family) and the FokI enzyme fused to zinc finger nuclease (ZFN) and Transcription Activator-like Effector Nuclease (TALEN) systems produce a staggered DSB with a 5′ overhang of 4 or 5 nucleotides[12–15]. Focused screening of 476 genes known to be involved in DNA repair coupled with short-read sequencing has revealed how genetic alterations can influence local DSB repair outcomes stemming from either a Cas9 or Cas12a DSB[16]. However, larger-scale effects and the role of the remaining 98% of the protein-encoding genome remain to be uncovered.

We used genome-wide CRISPR interference (CRISPRi) screening to interrogate the genetic dependencies of differential DSB architectures on repair after introducing a single Cas9 or Cas12a break into the genome of human K-562 cells. This approach identified known and novel factors required for the specific repair of blunt or overhanging DSBs. We focused on ERCC6L2, a poorly characterized ATPase that we found played a minimal role in Cas9-induced DSB repair but was crucial for normal repair of Cas12a-induced DSBs. *ERCC6L2* knockout, ATPase-mutant, and patient-derived cells normally repaired Cas9 DSBs to yield short insertions and deletions. However, these cells failed to properly repair Cas12a DSBs, resulting in large genomic deletions and translocations. Multiple Cas12a DSBs in *ERCC6L2* knockout and patient-derived cells resulted in cellular toxicity, as did multiple staggered DSBs generated by etoposide-induced TOP2 trapping. Mechanistically, we found that ERCC6L2 limits ATM-induced end-resection mediated by the MRE11-RAD50-NBS1 (MRN) complex to promote efficient NHEJ by directly binding and melting overhanging DNA ends. Collectively, our data explain differential reports of the importance of ERCC6L2 for NHEJ, which retrospectively differ in the molecular architecture of the induced DSB.

## Results

### ERCC6L2 is selectively required for repair of Cas12a-induced staggered DSBs

To elucidate factors underpinning cellular responses linked to different genome editor-induced DSB architectures, we performed genome-wide CRISPRi dropout screens. We engineered a modified mCherry construct ("CasCherry") that contains protospacer adjacent motifs (PAMs) of Cas9 (NGG) and AsCas12a (TTTV) in close proximity and was used to track transduction and editing efficiency (Supplementary Fig. 1a). This reporter was installed on a lentiviral cassette containing a genome-wide CRISPRi guide RNA (gRNA) library and transduced into

clonal CRISPRi K-562 dCas9-BFP-KOX1 cells at low multiplicity of infection to yield one gRNA per cell (Fig. 1a). These cells were electroporated with Cas9 ribonucleoprotein (RNP; Cas protein complexed to synthetic gRNA) to generate a blunt DSB, Cas12a RNP to create a 5′ staggered overhang DSB, or a mock plasmid control. After cells had been propagated for two weeks, their genomic DNA was extracted and used to generate targeted next generation sequencing (NGS) libraries of the gRNA cassette. NGS revealed gRNAs that were depleted or enriched in each of the three cell populations (Supplementary Fig. 1b–f).

In comparing Cas12a-treated cells to Cas9-treated cells, we observed that genes for several known NHEJ factors, such as *LIG4*, *NHEJ1*, and *PRKDC* were more important for a blunt Cas9 DSB than a staggered Cas12a DSB (Fig. 1b). By contrast, genes selectively depleted in Cas12a-treated cells included the MMEJ factor *POLQ* and factors connected to DSB processing and repair, such as *DCLRE1C* (Artemis), *NBN* (NBS1), and *ERCC6L2*[8,17,18]. ERCC6L2 (DNA excision repair protein ERCC-6-like 2) is a SWI/SNF-like ATPase that functionally clusters with NHEJ factors in some genetic screens[19–21]. It facilitates optimal class switch recombination (CSR) in mice[18–20] and restricts DNA end resection[22]. However, *ERCC6L2* dependence in NHEJ has also been found to vary across different studies, raising questions about its importance for NHEJ processes[18]. In human subjects, bi-allelic mutations of *ERCC6L2* cause inherited bone marrow failure (BMF) syndrome, myelodysplastic syndrome, leukemia, and neurological disorders[23,24], but little is known about how *ERCC6L2* mutations yield these aetiologies. Crucially, no current model of ERCC6L2 function explains the striking differential hypersensitivities of ERCC6L2 deficient cells to Cas12a vs Cas9 DSBs highlighted by our screen data. We therefore investigated the potential role of ERCC6L2 in the repair of staggered and blunt DSBs.

We first individually assessed whether loss of ERCC6L2 was toxic to cells where a Cas12a DSB was introduced at a single locus. By using a flow cytometry-based growth competition assay with two *ERCC6L2* targeting gRNAs (Fig. 1c), we observed in two distinct cell types (human K-562 and RPE1) that ERCC6L2 knockdown (KD) did not significantly increase the toxicity of a Cas12a DSB (Fig. 1d, Supplementary Fig. 1g). This led us to hypothesize that the apparent dropout of *ERCC6L2* gRNAs in the CRISPR screen might instead have been due to large genomic deletions that did not affect viability but caused loss of the gRNA cassette located 2 kb from the Cas target sites in the lentiviral cassette (Fig. 1a).

To ascertain if ERCC6L2 KD results in large deletions upon induction of Cas-induced DSBs, we targeted the *TRAC* locus with either Cas9 or Cas12a. *TRAC* is known to be prone to large deletions after a Cas DSB, giving us a positive background of large deletions in which to assess the additional effect of ERCC6L2 loss[25]. We measured editing outcomes by both short-read (Illumina) and long-read (Oxford Nanopore) NGS (Fig. 1e). These assays accurately measure short (1 to 100 bp) and large (kilobase) deletions, respectively. ERCC6L2 had no observable effect on either short or large deletions stemming from a blunt Cas9 DSB (Fig. 1f–h). By contrast, unlike the situation with Cas9, a staggered Cas12a DSB at *TRAC* combined with ERCC6L2 KD caused a marked reduction of smaller indels (<10 bp) and a substantial increase in larger indels as measured by both short read (11–99 bp) (Fig. 1f) and long read (1 kb+) sequencing (Fig. 1g, h).

### ERCC6L2 prevents large deletions and translocations during staggered DNA break repair

We next tested whether ERCC6L2 generally played a role in modulating deletion lengths at staggered Cas12a DSBs but not blunt Cas9 DSBs. We used CRISPR-Cas to generate multiple *ERCC6L2* knockout clones in a human RPE1 *TP53*−/− cell background (Supplementary Fig. 2a) and ensured *ERCC6L2* knockout did not affect cell cycling or

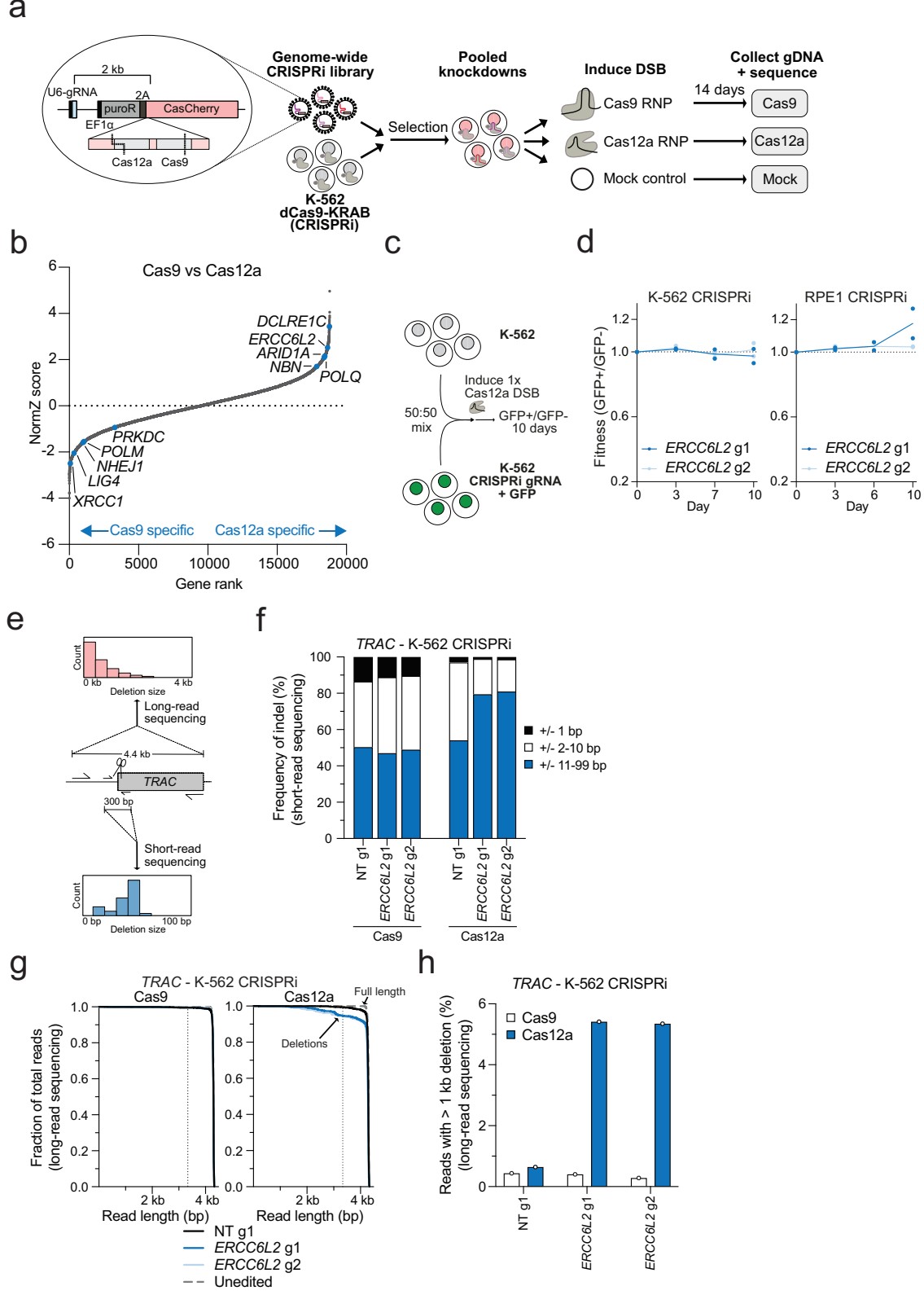

cell proliferation in this cell line (Supplementary Fig. 2b, c). We then induced DSBs in these cells with Cas9 or Cas12a at three different genomic loci (*TRAC*, *CCR5*, and *AAVS1*) and measured indel outcomes using both short and long read sequencing. In all cases, we observed minimal differences in the indel profiles resulting from a Cas9 DSB between *ERCC6L2*$^{-/-}$ and wild-type clones. However, compared to wild-type cells, a Cas12a DSB resulted in larger indels and a marked

increase in long deletions in the *ERCC6L2*$^{-/-}$ clones (Fig. 2a, b and Supplementary Fig. 2d–m).

Since mutation of *ERCC6L2* is pathogenic in humans, leading to bone marrow failure and hematological malignancies, we explored whether a Cas12a-induced DSB also specifically yields large deletions in cells derived from such individuals. We used a Cas9 or Cas12a RNP to target *CCR5* in bone marrow stromal cells (BMSCs) obtained from a

**Fig. 1 | A genome-wide screen implicates *ERCC6L2* in the repair of Cas12a-induced damage. a** Workflow for a dropout-based CRISPR inhibition screen to discern hypersensitivities to Cas9 (blunt double-strand break (DSB)) or Cas12a (5′ staggered DSB) editing. The Weissman v2[66] genome-wide gRNA library contains a modified mCherry ("CasCherry") engineered with neighboring nuclease target sites to track transduction and editing efficiency. The gRNA cassette is located 2 kb upstream from the nuclease target sites. **b** Aggregated screen output results comparing gene-level normalized Z (normZ) effect scores of knocked-down genes in Cas12a relative to Cas9 edited populations (Cas12a gRNA counts/Cas9 gRNA counts). Genes selectively depleted in Cas12a edited cells have a positive normZ effect score while genes selectively depleted in Cas9 edited cells have a negative normZ effect score. Notable DNA repair genes are annotated. **c** Scheme for a fluorescence-based competition assay to validate candidate genes. A dDSB was introduced using the CasCherry targeting gRNA using Cas12a in a 50:50 mixed population of wild-type and ERCC6L2 knockdown (KD) cells. **d** Competition assay results in K-562 and RPE1 cell types for two different CRISPRi gRNAs targeting *ERCC6L2*. Lines represent the mean ($n = 2$ biological replicates). **e** Scheme for assessing editing outcomes at the genomic level. A single DSB was introduced with a Cas RNP at the 5′ end of *TRAC* and the surrounding region was PCR amplified with two different primer sets and sequenced using either short-read (Illumina) or long-read (Oxford Nanopore) sequencing technologies. **f** Editing outcomes binned by indel size based on short-read sequencing ($n = 1$ biological replicate). NT, non-targeting. **g** Cumulative read depth based on read length from long-read sequencing ($n = 1$ biological replicate). Deletions lead to a leftward shift of the curve. The dashed line corresponds to a 1 kb deletion length. **h** Quantification of the number of reads in long-read sequencing containing greater than a 1 kb deletion from (**g**). Source data are provided as a Source Data file.

healthy donor and a donor with a homozygous *ERCC6L2* mutation (NM_020207.4:c.2101 C > T, NP_064592.2:p.Gln701X). The indel profile for Cas9 was similar across genotypes, but Cas12a editing induced large indels and long deletions exclusively in the *ERCC6L2* mutant patient-derived BMSCs (Fig. 2c, d and Supplementary Fig. 2n, o).

As ERCC6L2 is a SWI/SNF family member, we sought to test whether its catalytic activity is required for its differential importance at Cas9 and Cas12a DSBs. Thus, we used CRISPR-Cas homology directed repair (HDR) to engineer an inactivating mutation (K154R) in the ATP binding pocket of the ATPase domain at the endogenous *ERCC6L2* locus[22] (Supplementary Fig. 3a). We isolated two homozygous, isogenic RPE1 *TP53*[−/−]; *ERCC6L2*[K154R] clones (Supplementary Fig. 3b) and used droplet digital PCR (ddPCR) to ensure biallelic installation of the desired mutation (Supplementary Fig. 3c). Compared to wild-type cells, the endogenous ATPase-null mutants exhibited increased large deletions in response to a Cas12a DSB but not a Cas9 DSB, analogous to the *ERCC6L2* knockout clones (Fig. 2e and Supplementary Fig. 3d–f). Hence, we concluded that ERCC6L2 ATPase activity restricts the formation of large deletions during the repair of staggered DSBs.

To orthogonally confirm the observed sequence loss phenotype is specific to ERCC6L2, we complemented an RPE *TP53*[−/−]; *ERCC6L2*[−/−] clone with inducible lentiviral expression of wild-type and K154R ERCC6L2 (Supplementary Fig. 3g). Overexpression of wild-type ERCC6L2 rescued large deletions back to wild-type levels, while overexpression of the K154R mutant only partially rescued the phenotype (Fig. 2f and Supplementary Fig. 3h–j). We therefore concluded that ERCC6L2 catalytic activity has a strong contribution to its effects on staggered-end DSB repair.

Next, we explored whether ERCC6L2 generally acts at staggered DSBs or is somehow specific for Cas12a-generated DSBs. We first used a FokI TALEN to generate DSBs with 5′ overhangs. Using two different TALEN target sites at the *TRAC* locus, *ERCC6L2*[−/−] cells again displayed a decrease in insertions and smaller indels compared to wild-type controls (Supplementary Fig. 4a–c). We then employed a CRISPR-Cas9 dual nickase "PAM-out" strategy to generate staggered DSBs of differing polarities. By using the same paired guide RNAs and only changing the Cas9 nickase, we created 5′ overhangs (Cas9 D10A) or 3′ (Cas9 H840A) overhangs at two different loci (*TRAC* site 1 and 2) (Fig. 2g). At both tested sites, ERCC6L2 loss increased 1 kb+ deletions for both 5′ and 3′ overhangs (Fig. 2h, i and Supplementary Fig. 4d–g). Overall, our data indicated that ERCC6L2 is necessary to prevent loss of genetic information for a wide variety of staggered, nuclease-induced DSBs.

Cas-induced DSBs can result in large-scale genomic alterations such as translocations and chromosome arm loss[26–28]. We used CAST-seq to determine if ERCC6L2 plays a role in translocations during Cas-induced DSB repair[28]. In Cas9-edited cells, ERCC6L2 status had no apparent effect on the number of genomic aberrations, including translocations (Fig. 2j and Supplementary Fig. 4h, i). In stark contrast,

cells edited with Cas12a exhibited a substantial increase in the abundance of translocations, from none observed in wild-type cells to ten unique translocation sites in *ERCC6L2*[−/−] cells (Fig. 2j and Supplementary Fig. 4h, i). To quantify very large deletions and chromosome arm loss, we monitored the loss of a uniquely mapped single copy GFP reporter after inducing a Cas DSB 1 Mb upstream (Fig. 2k)[29]. In agreement with the CAST-seq data, ERCC6L2 loss resulted in a significant 4-fold loss of GFP signal only in Cas12a edited cells (Fig. 2k). Considering all these data, we concluded that ERCC6L2 protects cells not only from deletions but also more large-scale outcomes such as translocations and chromosome arm loss during staggered DSB repair.

## ERCC6L2 protects against staggered DSB-induced cell toxicity and excessive end resection

Our data so far indicated that ERCC6L2 prevents sequence loss and/or rearrangement of genomic sequence during staggered DSB repair at single break sites. We next explored whether compounding such large-scale genomic damage negatively affected cell viability. Accordingly, we simultaneously introduced multiple staggered DSBs with a promiscuous Cas12a gRNA that matches 28 predicted identical sites dispersed across the genome ("gMulti") without directly targeting any essential gene (Supplementary Fig. 5a, b). Using DISCOVER-seq (MRE11 ChIP-seq)[30], we validated that 25 of the 28 predicted perfectly-matched on-target sites and two additional off-target sites contained distinctive MRE11 peaks indicative of active DSB repair (Supplementary Fig. 5a, c, d). We confirmed the efficiency of gMulti at one of these targeted sites by short-read NGS (Supplementary Fig. 5e). In a cell growth competition assay[31] (Fig. 3a), we observed that gMulti induced selective toxicity in *ERCC6L2*[−/−] cells, resulting in a ~60% loss relative to wild-type cells (Fig. 3b). To elucidate the growth defect caused by multiple DSBs in *ERCC6L2*[−/−] cells, we microscopically inspected cells for micronuclei which arise from unrepaired DNA damage that eventually leads to missegregation of chromosomes or acentric chromosomal fragments in mitosis[32]. Editing with gMulti significantly increased the number of *ERCC6L2*[−/−] cells harboring micronuclei ($64.8 \pm 11.6\%$) compared to wild-type cells ($33.3 \pm 10.8\%$) (Fig. 3c, d).

To complement the above studies, we next employed gMulti in patient BMSCs and again observed a significant decrease in cell viability only in the *ERCC6L2*[−/−] donor edited with gMulti (Fig. 3e and Supplementary Fig. 5f, g). We performed annexin V staining to mark apoptotic cells and observed that an average of 44% of gMulti edited *ERCC6L2*[−/−] BMSCs stained positive for apoptosis (Fig. 3f and Supplementary Fig. 5h). This was in sharp contrast to the healthy donor cells, where only 19% of cells were apoptotic. We concluded that by introducing multiple staggered DSBs with gMulti, we produced an increased burden of large genomic aberrations in *ERCC6L2*[−/−] cells that impaired cell proliferation and led to apoptotic death.

To test the role of ERCC6L2 in a context beyond genome editing and gain insights into its normal physiological function, we examined endogenous DSBs with staggered ends that arise during normal DNA

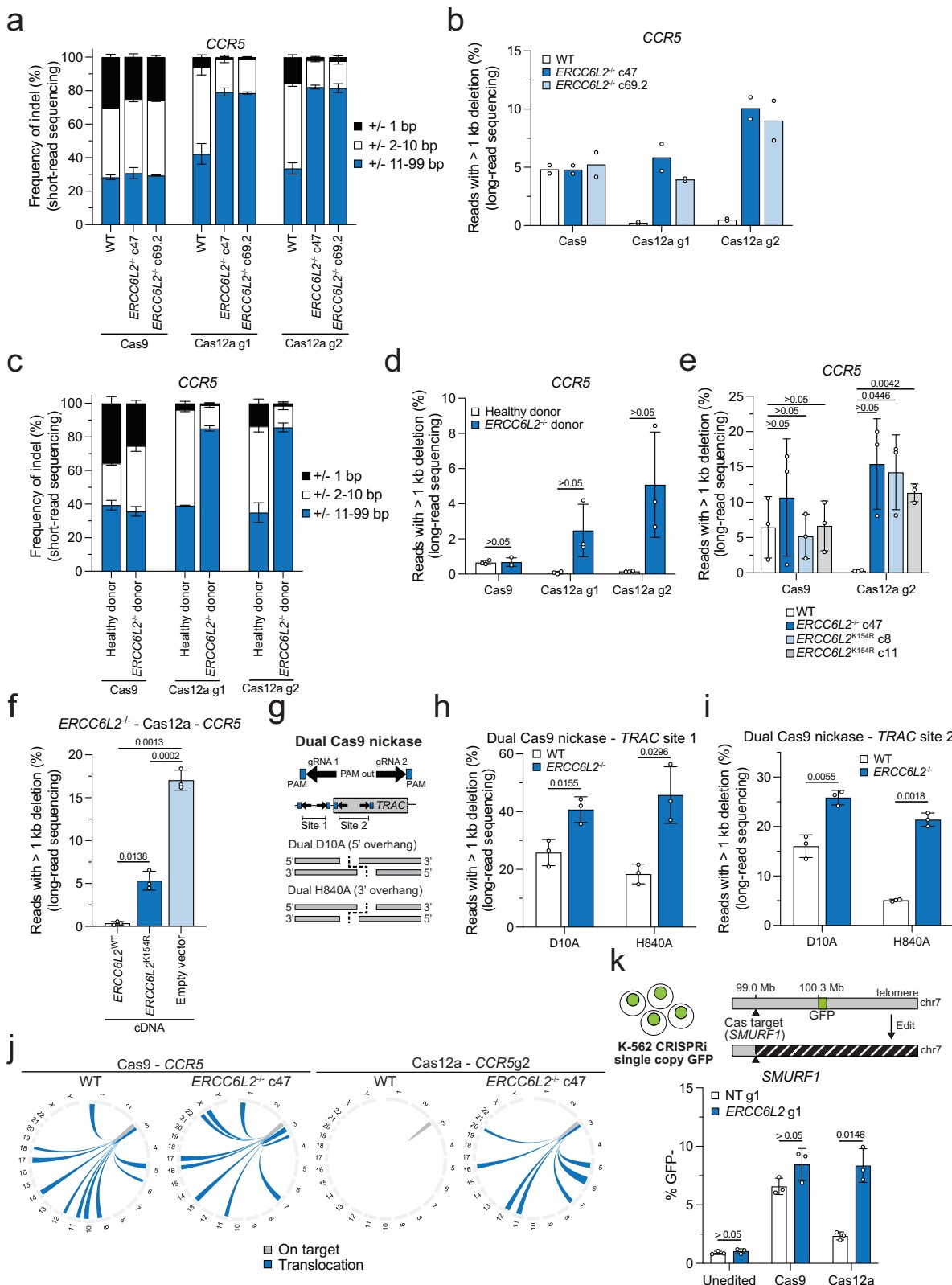

metabolism. DNA topoisomerase II (TOP2) modulates DNA topology by introducing a transient DSB harboring 4-bp cohesive 5′-overhangs (Fig. 3g and Supplementary Fig. 6a). Under normal circumstances, this cleavage is a transient intermediate, wherein the 5′ moiety of each DNA end is covalently attached to TOP2 by a phospho-tyrosyl linkage and quickly resealed. However, exposure to the chemotherapeutic drug etoposide or the presence of endogenously arising DNA lesions in

close proximity can stabilize the cleavage intermediate, thereby leading to enzymatic processing events that yield a DSB with staggered ends[33]. We induced TOP2 DSBs by etoposide treatment of p53 (TP53) proficient and p53 deficient cells that were also either ERCC6L2 wild-type or knockout. As with multi-targeting Cas12a, we found that ERCC6L2[−/−] cells exhibited significantly elevated etoposide hypersensitivity relative to wild-type cells (Fig. 3h), an impact that was

**Fig. 2 | ERCC6L2 loss results in large chromosomal aberrations upon genome editor induced staggered DSBs. a** Indel outcomes are binned by size based on short-read sequencing in RPE1 *TP53*$^{-/-}$ cells ($n$ = 3 biological replicates) WT, wild-type. **b** Quantification of reads in long-read sequencing containing greater than a 1 kb deletion in RPE1 *TP53*$^{-/-}$ cells ($n$ = 2 biological replicates). **c** Indel outcomes are binned by size based on short-read sequencing in healthy donor or patient-derived bone marrow stromal cells (BMSCs) ($n$ = 3 biological replicates). **d** Quantification of >1 kb deletions in BMSCs ($n$ = 3 biological replicates). **e** Quantification of >1 kb deletions in ATPase deficient *ERCC6L2*$^{K154R}$; *TP53*$^{-/-}$ RPE1 cells ($n$ = 3 biological replicates). **f** Quantification of >1 kb deletions in doxycycline inducible complementation of ERCC6L2 in RPE1 *TP53*$^{-/-}$; *ERCC6L2*$^{-/-}$ cells ($n$ = 3 biological replicates) **g** Scheme depicting how pairs of dual Cas9 D10A or H840A nickases are employed in a PAM-out orientation to generate 5′ or 3′ staggered DSBs,

respectively. Two unique gRNA pairs target *TRAC* site 1 (39 bp apart) or *TRAC* site 2 (54 bp apart). **h, i** Quantification of reads with deletion lengths greater than 1 kb in dual-nickase edited RPE1 *TP53*$^{-/-}$ cells at *TRAC* site 1 (**h**) and *TRAC* site 2 (**i**) ($n$ = 3 biological replicates). **j** Analysis of CAST-seq detected translocations resulting from genome editing in RPE1 *TP53*$^{-/-}$ cells ($n$ = 2 technical replicates). The gray line depicts the onsite site for *CCR5* on chromosome 3 and the blue lines indicate observed translocations between the *CCR5* target site and a distal genomic site. **k** Quantification of GFP loss as a proxy for 1 Mb+ deletions and chromosome arm loss in K-562 CRISPRi cells expressing the indicated gRNA measured 7 days post-editing ($n$ = 3 biological replicates). Bar graphs are displayed as mean ± SD. Statistical significance was determined with a two-tailed Welch's t-test in (**d, e, f, h, i, k**). Source data are provided as a Source Data file.

independent of p53 status (Supplementary Fig. 6b, c). Importantly, this effect was rescued by complementation with full-length ERCC6L2 but not the K154R ATPase mutant (Fig. 3i and Supplementary Fig. 6d).

DNA DSBs can be resected to single-stranded DNA (ssDNA) ends during break repair, and excessive or unresolved resection can lead to sequence loss reminiscent of the large deletions we observed by long-read sequencing. Unlike targeted Cas genome editing, etoposide treatment yields damage that is impossible to uniquely map on the sequence level[33]. We therefore assessed DNA-end resection in etoposide-treated cells by performing flow cytometry and quantifying accumulation of the single-stranded DNA-binding protein RPA in cells positive for γH2AX staining, an early marker of DSB induction[34]. ERCC6L2 loss caused a marked increase in RPA/γH2AX positive cells after 1 h of etoposide treatment (Supplementary Fig. 6e, f). This increase was evident during G2 phase of the cell cycle, where ERCC6L2 deficiency resulted in a nearly 3-fold increase in RPA/γH2AX positive cells (Supplementary Fig. 6e, f). Elevated RPA recruitment upon ERCC6L2 loss was independent of p53 status (Supplementary Fig. 6g, h).

To determine whether this increase in RPA staining in ERCC6L2-null cells reflected more DSBs undergoing resection and/or more extensive resection at individual DSBs, we monitored RPA foci formation at various times after acute etoposide treatment, during which DNA repair was taking place. Notably, 3 and 4 h after etoposide withdrawal, compared to control cells, *ERCC6L2*$^{-/-}$ cells showed both a greater number of RPA foci co-localizing with γH2AX in both S and G2 phases and an increased intensity of RPA foci (Fig. 3j, k and Supplementary Fig. 6i, j). As the intensity of RPA foci increases proportionally with the distance travelled by resection nucleases[35], this implies that ERCC6L2 deficiency promotes an increase in resection length. When we measured formation of bromodeoxyuridine (BrdU) foci under native (non-denaturing) conditions as an orthogonal measure of ssDNA generated by resection following etoposide treatment, we also observed higher levels in *ERCC6L2*$^{-/-}$ cells than in wild-type controls (Supplementary Fig. 6k).

To experimentally quantify the extent of DNA resection, we employed the Single Molecule Analysis of Resection Tracks (SMART) assay (Fig. 3l)[36]. SMART relies on a modified DNA combing technique to measure resection progression at the level of individual DNA fibers. We found that following etoposide treatment, *ERCC6L2*$^{-/-}$ cells had a median resection length of 21.45 μm compared to 16.64 μm in wild-type cells (Fig. 3m). These results thereby established that ERCC6L2 loss generated similar phenotypes in the contexts of both TOP2- and Cas12a-induced DSBs. When we used the DSB Inducible via AsiSI (DIVA) assay—a further approach to measure ssDNA lengths[37,38]—we demonstrated increased ssDNA 200 bp from the AsiSI cut site in *ERCC6L2*$^{-/-}$ cells compared to wild-type cells (Supplementary Fig. 6l, m). As AsiSI generates only a 2 bp overhang, this points to ERCC6L2 having impacts even when DSB overhang lengths are minimal. We note, however, that these resection experiments do not have a blunt DSB control. In summary, our data indicated that ERCC6L2 is involved in repairing

staggered DSBs, promoting cell fitness by preventing excessive end resection, sequence loss, and chromosomal translocations.

## Functional relationships between ERCC6L2 and ATM dependent resection mediated by the MRN complex

ERCC6L2 has previously been described as a putative DNA end joining factor[18,19,22]. To understand how large deletions are generated after loss of ERCC6L2, we used small molecules to inhibit factors involved in both canonical NHEJ and MMEJ. As expected, inhibition of the catalytic subunit of DNA-dependent protein kinase (DNA-PKcs) in a wild-type background led to increased sequence loss upon either Cas9 or Cas12a editing (Supplementary Fig. 7a–c). The absence of ERCC6L2 further exacerbated these deletions (Supplementary Fig. 7a–c) but did not increase the chromosome arm loss caused by DNA-PKcs inhibition (Supplementary Fig. 7d, e). This suggested that ERCC6L2 loss promotes kilobase-scale deletions independently of canonical NHEJ inhibition. By contrast, inhibition of POLθ, the primary polymerase involved with MMEJ, rescued kilobase-scale deletions under all conditions (Supplementary Fig. 7a–c). POLθ inhibition did not rescue megabase-scale deletions and chromosome arm loss in ERCC6L2 KD cells (Supplementary Fig. 7d, e). This demonstrated that in the absence of ERCC6L2, kilobase-scale deletion events are in part dependent on MMEJ and that multiple pathways are involved in repairing the DNA damage.

The DNA-damage activated protein kinase ATM regulates both initial and long-range DSB end resection by phosphorylating CtIP, the MRN (MRE11-RAD50-NBS1) complex, BLM, and EXO1 (reviewed in ref. 39). Given the effects of ERCC6L2 on end resection, we next explored its potential functional relationship with ATM. We generated *ATM*$^{-/-}$ clones and infected them with lentivirus expressing gRNAs targeting either *LacZ* as a control or *ERCC6L2* (Supplementary Fig. 8a–c). By using cell-growth competition assays, we found that ERCC6L2 loss did not increase the etoposide hypersensitivity of *ATM*$^{-/-}$ cells (Fig. 4a), a result that was also observed in clonogenic cell survival assays (Supplementary Fig. 8d). This implied that the role of ERCC6L2 in response to etoposide-induced DNA damage is no longer required in the absence of ATM. To corroborate these findings, we made use of the multi-targeting Cas12a gMulti and treated cells with DMSO or an ATM kinase inhibitor (ATMi; AZD0156). Notably, in comparison to DMSO treatment, ATM inhibition partially rescued the dropout of *ERCC6L2* knockout cells relative to wild-type cells, again, indicating a functional connection between ERCC6L2 and ATM dependent processes (Supplementary Fig. 8e).

To determine whether the extensive resection observed in *ERCC6L2*$^{-/-}$ cells depends on ATM, we measured RPA accumulation following acute etoposide and ATMi treatment. Strikingly, ATMi counteracted the elevated RPA accumulation observed in etoposide-treated ERCC6L2 deficient cells in G2, reducing it from 64% (± 17.4) to 18% (± 17.7) (Fig. 4b and Supplementary Fig. 8f). ATMi also counteracted etoposide-induced RPA accumulation in ERCC6L2 proficient cells specifically in G2, decreasing the percentage of RPA-positive cells from 28% (± 7.6) to 8% (± 6.5). Complementing these results, we found

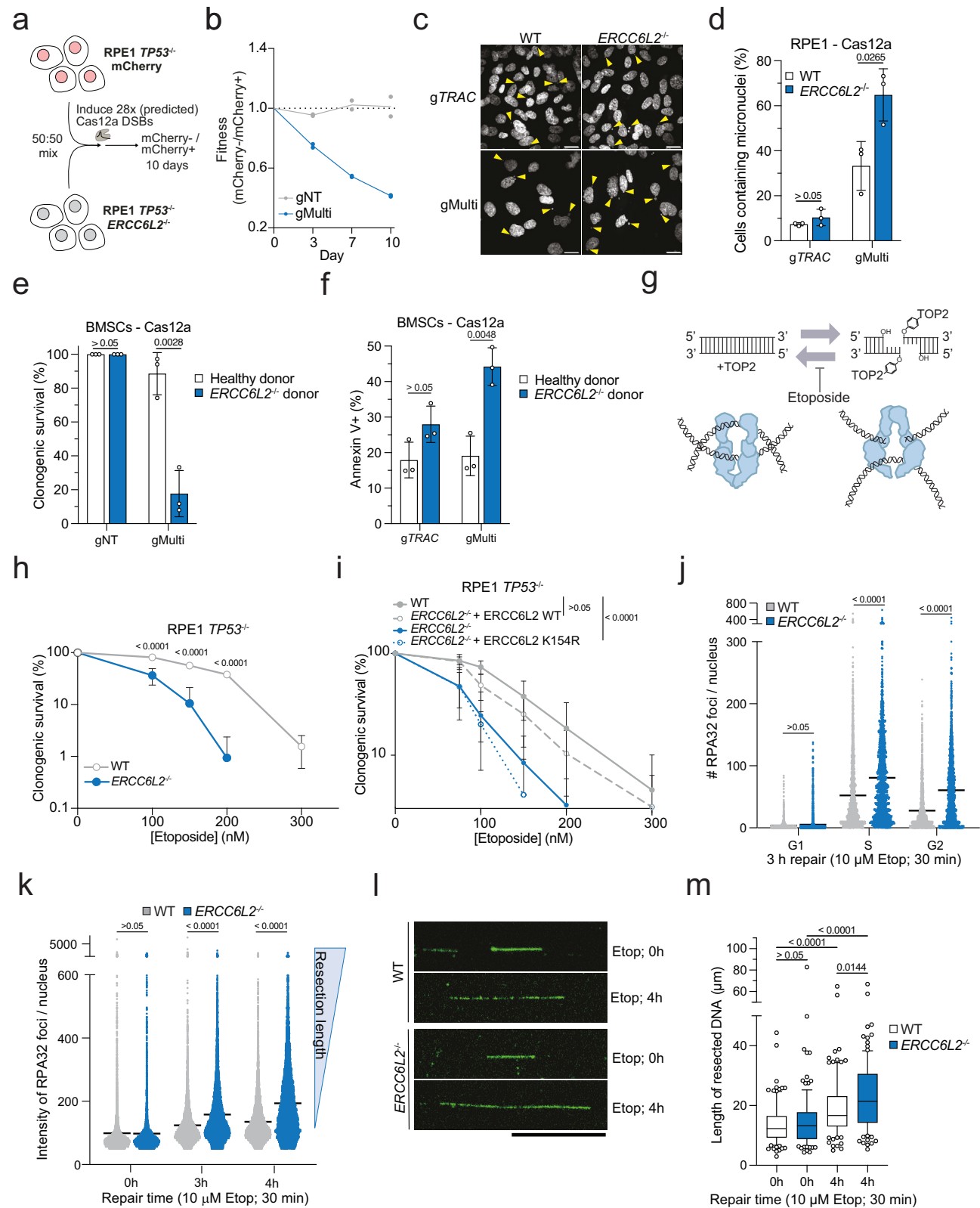

through DNA sequencing of Cas12a editing products that in both wild-type and *ERCC6L2*<sup>−/−</sup> cells, ATMi reduced the frequencies of Cas12a-induced large deletions (Supplementary Fig. 8g, h). Together, these findings suggested that ERCC6L2 counteracts ATM-driven resection following induction of DSBs with staggered ends.

ATM has more than 700 identified substrates[40]. To attempt to discern which factors are involved in ERCC6L2's role in preventing excessive resection and sequence loss after staggered DSB induction, we conducted genome-wide CRISPR-Cas9 screens in RPE1 *TP53*<sup>−/−</sup>; *ERCC6L2*<sup>−/−</sup> cells with and without etoposide treatment (Fig. 4c and Supplementary Fig. 9a). We compared our results to those obtained from an identical screen performed in an ERCC6L2 proficient background[41]. In ERCC6L2 proficient cells, the strongest candidate genes for etoposide sensitivity included *ERCC6L2* itself, as well as

**Fig. 3 | Multiple staggered DSBs manifest in cell toxicity and excessive resection. a** Scheme for fluorescence-based competition assay. **b** Quantification of cell fitness after editing with the indicated gRNAs (*n* = 2 biological replicates). **c** Representative fluorescence microscopy images of DAPI-stained RPE1 *TP53*⁻/⁻ cells 72 h post-Cas12a editing with the indicated gRNAs. Micronuclei are marked with yellow triangles. WT, wild-type. Scale bar = 10 μm. **d** Quantification of micronuclei 72 h post-treatment from (**c**) (mean of *n* = 3 biological replicates ± SD). Each data point corresponds to an individual biological replicate with >100 cells. **e** Quantification of clonogenic survival of bone marrow stromal cells (BMSCs) edited with a non-targeting gRNA (gNT) or the Cas12a gMulti (mean of *n* = 3 biological replicates ± SD). **f** Quantification of apoptotic BMSCs treated with Cas12a targeting either a single locus (g*TRAC*) or multiple targets (gMulti) (mean of *n* = 3 biological replicates ± SD). **g** Schematic of etoposide's mechanism of action. **h** Clonogenic survival assay of RPE1 *TP53*⁻/⁻ cells titrated with etoposide (mean of *n* = 3 biological replicates ± SD). **i** Clonogenic survival assay of RPE1 *TP53*⁻/⁻ cells complemented with full-length or K154R ERCC6L2 titrated with etoposide (mean of *n* = 3 biological replicates ± SD). **j** Quantification of the number of RPA32 foci in G1, S and G2 phases in RPE1 *TP53*⁻/⁻ cells (mean of *n* = 3 biological replicates). **k** Intensity quantification of RPA32 foci co-localizing with γH2AX foci in RPE1 *TP53*⁻/⁻ cells (mean of *n* = 3 biological replicates). **l** Representative SMART assay tracts. Scale bar = 20 μm. **m** Quantification of resected DNA tracts measured. Box plots show the median and interquartile range (box), with whiskers extending to the 10th and 90th percentiles. At least 100 DNA fibers were measured per condition. Data from one representative experiment are shown; the experiment was performed two independent times with similar results. Statistical significance was determined with a two-tailed Welch's t-test in (**d, e, f, m**); two-way ANOVA and Šídák comparison test in (**h**); two-way ANOVA and Tukey's multiple comparison test in (**i**); one-way ANOVA and Tukey's multiple comparison test in (**j, k**). Source data are provided as a Source Data file.

known factors involved in cellular responses to etoposide (e.g., *TDP2*, which removes the covalent link between TOP2 and DNA 5′ ends, and the broad-spectrum drug exporter *ABCC1*), key components of the NHEJ machinery (*NHEJ1*, *PRKDC*, and *LIG4*), and *ATM* (Fig. 4d and Supplementary Fig. 9b). ERCC6L2 deficient cells were hypersensitized to etoposide by loss of many of the same genes as wild-type cells. In agreement with previous Cas12a editing data (Supplementary Fig. 7a–e), neither loss of NHEJ factors nor POLθ rescued the etoposide hypersensitivity of *ERCC6L2*⁻/⁻ cells (Fig. 4d and Supplementary Fig. 9b). However, two genes had a strong differential ERCC6L2-dependent effect. First, *ERCC6L2* was not identified as a hit in *ERCC6L2*⁻/⁻ cells, confirming the robustness of the screen (Fig. 4d and Supplementary Fig. 9b). Secondly, loss of the *NBN* (NBS1) component of the MRN complex hypersensitized wild-type cells to etoposide but had essentially no effect in the absence of ERCC6L2 (Fig. 4d and Supplementary Fig. 9b).

Importantly, we validated the genetic interaction between *ERCC6L2* and *NBN* by infecting wild-type or *NBN*⁻/⁻ cells with a lentivirus expressing a gRNA targeting *ERCC6L2* (Supplementary Fig. 9c, d) and performing clonogenic survival assays. Indeed, while knockout of *ERCC6L2* conferred hypersensitivity to etoposide in wild-type cells, it did not increase the sensitivity of *NBN*⁻/⁻ cells, further highlighting epistasis between the two factors (Fig. 4e and Supplementary Fig. 9c, d). We also observed similar phenotypes with MRE11 and the MRN resection partner CtIP, where siRNA depletion of MRE11 or RBBP8 (CtIP) promoted clear hypersensitivity to etoposide in the wild-type background but had no marked effect in *ERCC6L2*⁻/⁻ cells (Fig. 4f and Supplementary Fig. 9e, f).

Like TDP2, the MRN complex can help remove covalent TOP2 blockages from 5′ DNA ends[42], potentially explaining why NBN or MRE11 loss harms wild-type cells upon etoposide treatment. However, MRN also plays an additional role in promoting resection to form ssDNA during DSB processing. Our data suggested that ERCC6L2 counteracts excessive end resection initiated by MRN activity. Indeed, loss of NBN completely suppressed the increased RPA foci formation after etoposide treatment in *ERCC6L2*⁻/⁻ cells (Fig. 4g and Supplementary Fig. 9c, d). Accordingly, chemical inhibition of MRE11 completely rescued Cas12a-induced large deletions in *ERCC6L2*⁻/⁻ cells (Fig. 4h and Supplementary Fig. 9g). Since MRN can both promote and impair DNA damage resolution, we speculate that its loss may shift from being harmful in wild-type cells to having a neutral or lesser effect in *ERCC6L2*⁻/⁻ cells where loss of DNA sequence is increased.

Given the functional interaction between ERCC6L2 and the MRN complex, we wondered if ERCC6L2 physically impaired MRN recruitment, retention, or footprint at DSBs. We therefore performed proximity ligation assays (PLA) with antibodies against NBS1 and γH2AX (Fig. 4i). We observed clear induction of proximity between the proteins in wild-type cells, with this induction being similar in ERCC6L2 deficient cells. This suggests that ERCC6L2 does not regulate the etoposide-induced recruitment of NBS1 to chromatin. ERCC6L2 status also did not affect the etoposide-induced interaction of MRE11 with CtIP as measured similarly by PLA (Fig. 4j). However, leveraging our ChIP-seq data (Supplementary Fig. 5c, d), we measured significantly increased MRE11 signal spreading surrounding Cas12a gMulti target sites in *ERCC6L2*⁻/⁻ cells relative to wild-type cells (Fig. 4k, l). This indicated that ERCC6L2 limits MRN spreading at DNA termini and prevents its function at more distal regions. Thus, we speculate that in the absence of ERCC6L2, MRN initiates more end-resection events that ultimately manifest as large deletions, chromosome aberrations, and cytotoxicity, particularly in contexts where a DSB is normally repaired by NHEJ.

## Recombinant ERCC6L2 specifically melts DNA substrates with overhangs

To define the physical DNA binding and potential end-melting properties of ERCC6L2, we expressed and purified recombinant human ERCC6L2 from insect *Sf*9 cells (Supplementary Fig. 10a). Through electrophoretic mobility shift assays (EMSAs) with varying DNA substrates (Fig. 5a), we found that ERCC6L2 had a high affinity for double-stranded DNA (dsDNA) and a somewhat lower affinity for ssDNA (Fig. 5b, c). We also established that ERCC6L2 effectively bound end architectures ranging from blunt ends to overhangs to Y-structures (Fig. 5b, c). We validated ERCC6L2's preferential binding of dsDNA over ssDNA by using competition assays with excess unlabeled DNA (Supplementary Fig. 10b–d). Despite ERCC6L2 binding all dsDNA substrates tested, we observed that it melted dsDNA substrates with either 5′ or 3′ four nucleotide overhangs but not blunt-ended DNA or Y-structured DNA (Fig. 5d, e). The DNA melting activity of ERCC6L2 was only observed in the presence of ssDNA binding (SSB) proteins RPA or human mitochondrial SSB (mtSSB), which likely prevented reannealing of the ssDNA products[43] (Supplementary Fig. 10e).

We found that ERCC6L2's helicase activity was required to melt DNAs with overhanging ends. First, addition of non-hydrolyzable AMP-PNP reduced, but did not eliminate, the formation of ssDNA products, while the chelation of Mg²⁺ cofactor with EDTA completely disrupted DNA melting (Fig. 5f, g). Second, substitution of the lysine in the ATP binding pocket of ERCC6L2 (K154A, Supplementary Fig. 3a) did not negatively affect DNA binding of ERCC6L2 (Supplementary Fig. 10f, g) but reduced the DNA melting activity to the level observed in the presence of AMP-PNP (Fig. 5h, i). DNA melting activity partially independent of ATP hydrolysis was also previously observed with the MRN complex[44]. Taken together, these data indicated that ERCC6L2 binds DNA ends regardless of their architecture but acts specifically to melt dsDNAs harboring short, staggered overhangs in an ATP-dependent manner. These biochemical data are consistent with our cellular data on the differential cellular effects of ERCC6L2 catalytic activity on blunt Cas9 and staggered Cas12a or TALEN-induced DSBs, as well as with our dual nickase data showing

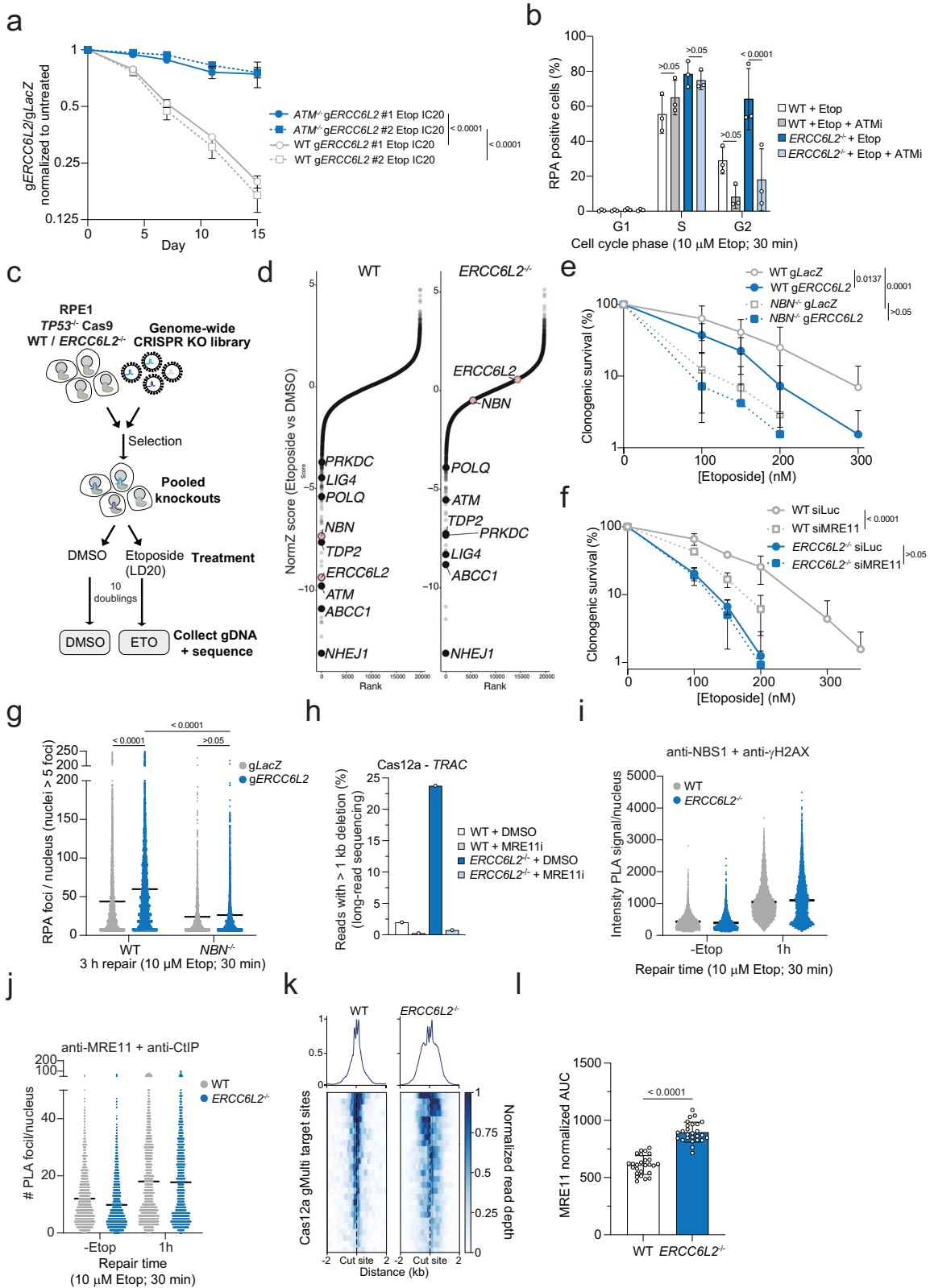

that ERCC6L2 exerts an effect independent of the polarity of the overhanging DNA ends (Fig. 2).

## Discussion

Our results demonstrate a crucial requirement for ERCC6L2 in the repair of staggered DSBs but not blunt DSBs and reveal the molecular consequences of ERCC6L2 loss. Upon generation of staggered DSBs,

ERCC6L2 prevents chromosomal translocations and excessive resection that would otherwise lead to large-scale genomic deletions. We propose a model wherein ERCC6L2 plays a specific and pivotal role in protecting staggered DSBs of both 5' and 3' polarities from mispairing and over-resection by melting them for prompt initiation of NHEJ (Fig. 5j). This aligns with several studies which have proposed that some degree of DNA-end unwinding stabilizes synapsis and promotes

**Fig. 4 | Physiological role of ERCC6L2 in avoiding loss of sequence.**
**a** Competition assay monitoring the ratio of RPE1 $TP53^{-/-}$ cells expressing gRNA targeting *LacZ* vs expressing gRNA targeting *ERCC6L2* after etoposide treatment (mean of $n = 5$ biological replicates ± SD). WT, wild-type. **b** Quantification of RPA32 positive RPE1 $TP53^{-/-}$ cells by cell cycle status. Cells were treated for 30 min with 10 nM ATMi (AZD0156), then cells were treated with 100 μM of etoposide + 10 nM ATMi for 1 h (mean of $n = 3$ biological replicates ± SD). **c** Scheme for a genome-wide CRISPR knockout screen in RPE1 $TP53^{-/-}$ (IC$_{20}$ = 120 nM, published in ref. [41]) and $TP53^{-/-}$; $ERCC6L2^{-/-}$ backgrounds (IC$_{20}$ = 45 nM). **d** Rank plots from normZ effect scores of etoposide vs DMSO of CRISPR knockout screens. **e** Clonogenic survival assay in RPE1 $TP53^{-/-}$ titrated with etoposide (mean of $n = 3$ biological replicates ± SD). **f** Clonogenic survival assay in RPE1 $TP53^{-/-}$ cells transfected with the indicated siRNAs and titrated with etoposide (mean of $n = 3$ biological replicates ± SD). **g** Quantification of RPA32 foci in RPE1 $TP53^{-/-}$ cells treated with etoposide. Nuclei

showing more than 5 RPA foci were selected (mean of $n = 3$ biological replicates). **h** Quantification of long-read sequencing reads with deletion lengths greater than 1 kb in RPE1 $TP53^{-/-}$ cells treated with either DMSO or MRE11 inhibitor (MRE11i) ($n = 1$ biological replicate). **i** Quantification of PLA signal intensity for NBS1-γH2AX in RPE1 $TP53^{-/-}$ cells treated with etoposide (mean of $n = 3$ biological replicates). **j** Quantification of PLA foci for MRE11-CtIP in etoposide-treated RPE1 $TP53^{-/-}$ cells (mean of $n = 2$ technical replicates). **k** MRE11 peak profile surrounding each Cas12a gMulti target site in RPE1 $TP53^{-/-}$ cells. **l** Normalized quantification of the area under the curve (AUC) of MRE11 at each of the 27 Cas12a gMulti target sites (1 biological replicate; mean ± SD). Statistical significance was determined by two-way ANOVA and Tukey's multiple comparison test in (**a, b, e, f**); one-way ANOVA and Tukey's multiple comparison test in (**g**); two-tailed Welch's t-test in (**l**). Source data are provided as a Source Data file.

DNA-PKcs activity during NHEJ[45–50]. In the absence of ERCC6L2 melting activity, NHEJ would thus be less efficient, which would provide greater opportunities for the MRN complex to initiate long resection through an ATM mediated process. This is consistent with ERCC6L2 recruitment to damage sites immediately after Ku and XLF, and prior to NBS1[18]. The increase in long range resection observed in ERCC6L2 deficient cells occurs mainly in S and G2 phases (Fig. 3j), likely due to regulation of CtIP expression and activity rather than a lack of ERCC6L2 activity in G1. Indeed, *ERCC6L2* KO cells also show an increase in small deletions upon the generation of Cas12-induced DSBs (Fig. 2a, c and Supplementary Fig. 2g, k), which could be due to the activity of the MRN complex on DSBs during G1 when they do not undergo extensive resection. How the opening of the overhang molecularly contributes to NHEJ remains unclear and might be resolved by future structural studies.

Our discovery that ERCC6L2 uniquely acts on overhanging DNA ends clarifies differential reports for the importance of ERCC6L2 at DSBs. Previous studies have proposed ERCC6L2 as a general NHEJ factor that is rapidly recruited to all DSBs to restrict DNA end resection[18,19,22]. But contrary to what one would expect from a general NHEJ factor, ERCC6L2 is dispensable for the repair of some DSBs and required for others. First, ERCC6L2 is reported to have little effect on DSBs induced by ionizing radiation (IR)[19] but is required to repair DSBs induced by etoposide (ref. [19] and this work). Our findings reconcile this apparent discrepancy. IR generates lesions with heterogeneous end architectures[51], diluting the effect of ERCC6L2 loss. In contrast, etoposide generates a uniform DSB end architecture with four nucleotide overhangs. Second, ERCC6L2 is required for successful orientation-specific joining during AID-induced CSR stemming from offset nicks[52]. However, ERCC6L2 was surprisingly not as relevant for Cas9-induced CSR[18]. Third, ERCC6L2 is also dispensable for the repair of blunt/hairpin DSBs induced during V(D)J recombination[18]. Our data mechanistically explain these seemingly conflicting observations by revealing that ERCC6L2 shows DNA-end melting activity and has cellular activity only at staggered DSBs.

Integrating across multiple DSB-inducing modalities, our data suggest that any staggered DSB requires ERCC6L2 action. While Cas9 typically produces a blunt DSB, it can also produce a staggered 1 bp overhanging DSB in certain sequence contexts[53]. Cas9 at the *AAVS1* locus produced a dominant 1 bp insertion that is likely due to a staggered 1 bp DSB (Supplementary Fig. 2k). At this site, we observed a minor but consistent loss of insertions in *ERCC6L2*$^{-/-}$ cells, a phenotype consistent with other Cas9 1 bp staggered sites[54]. Inducing 2 bp overhanging ends with AsiSI (Supplementary Fig. 6l, m) also resulted in increased ssDNA tracks in ERCC6L2 deficient cells. Conversely, our dual nickase strategy employed nicks up to 54 bp apart that yielded an ERCC6L2 dependence (Fig. 2i). Taken together, these data suggest that ERCC6L2 has the potential to act on any DSB overhang, including ones as small as 1 bp.

Human ERCC6L2 deficiencies manifest mainly as inherited BMF and leukemia[23,24]. Fanconi anemia (FA), the most common BMF syndrome[55], is typically caused by aldehyde-induced DNA interstrand crosslinks. However, *ERCC6L2* mutant cells are not hypersensitive to crosslinking agents such as mitomycin C[56], suggesting a different source of damage responsible for BMF associated with ERCC6L2 loss. TOP2 is a major source of endogenous staggered breaks, and we find that the ERCC6L2 loss confers hypersensitivity to TOP2-induced damage. Such lesions can accumulate at the promoters of essential genes during neuronal activation[57], suggesting a mechanism for aetiology of neurological symptoms in some ERCC6L2 patients[58,59]. Additionally, leukemia development has been extensively linked to TOP2 damage, with a significant proportion of etoposide-treated patients developing therapy-related secondary leukemias[60,61]. Our findings suggest a role for TOP2 activity in causing leukemia in clinical contexts of ERCC6L2 deficiency and provide clues to the molecular source of pathology in ERCC6L2 patients that could help guide potential treatments.

We have found that ERCC6L2 status profoundly affects the outcomes of Cas12a genome editing. Importantly, we note that attempting to cure an *ERCC6L2* mutant patient with Cas12a would likely generate a population of large deletions at the targeted locus that could prove deleterious. Our findings thus add to a growing body of literature highlighting the need for patient biomarkers for optimal genome correction therapies. For example, patients with FA deficiencies are unable to effectively perform HDR[62], and so base- or prime-editing treatments have been proposed as alternative therapeutic approaches[63]. We suggest that any editing modality that induces a staggered DSB (ZFNs, TALENs, type V Cas, and more) should be avoided when exploring potential genome editing approaches to treat patients with *ERCC6L2* mutations. This proscription might also be warranted for other genes identified in our initial screen for differential effects of Cas9 vs. Cas12a (e.g., *DCLRE1C* or *ARID1A*), though this remains to be tested.

Overall, our study demonstrates that ERCC6L2 specifically acts on staggered DNA ends to promote non-mutagenic repair. Repair of staggered DSBs in the absence of ERCC6L2 helicase activity induces chromosomal translocations and results in catastrophic sequence loss. These results shed light on molecular and genetic determinants of genome maintenance and highlight the importance of precision application of genome editing reagents based on a patient's genotype.

## Methods

### Nucleotide sequences

Guide RNA was purchased from Synthego and Integrated DNA Technologies (IDT). DNA was purchased from IDT and Microsynth AG. All guide RNA and DNA oligonucleotide sequences can be found in Supplementary Tables 1 and 2. Cloning of plasmids used in this study were performed using either standard ligation-based or Gibson assembly techniques.

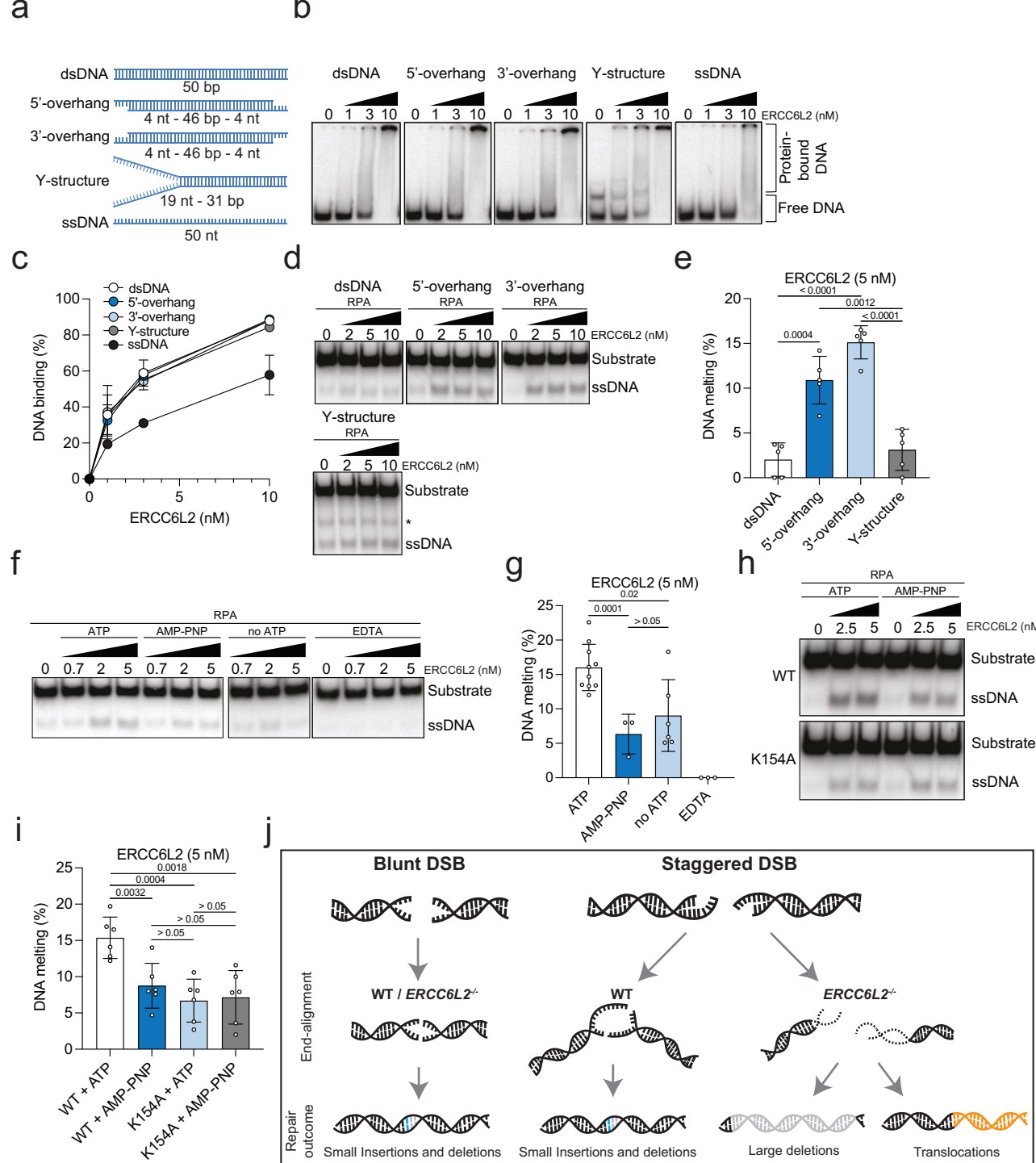

**Fig. 5 | Recombinant ERCC6L2 binds and melts DNA substrates with short overhangs. a** Schematic of the different DNA substrates for binding and melting assays. dsDNA, double-stranded DNA; ssDNA, single-stranded DNA; bp, base pair; nt, nucleotide. **b** Electrophoretic mobility shift assay (EMSA) with increasing concentrations of ERCC6L2 with the substrates depicted in (**a**). Protein-bound DNA exhibits an upwards gel shift. **c** Quantification of protein-bound DNA from (**b**) ($n = 3$ biological replicates). **d** DNA melting assay with increasing concentrations of ERCC6L2 in the presence of RPA to prevent reannealing of DNA. Asterisk denotes a substrate truncation species in the Y-structure DNA condition. **e.** Quantification of single-stranded DNA (ssDNA) product from (**d**) ($n = 5$ biological replicates). **f** DNA

melting assays of the 3' overhang substrate to assess activity of ERCC6L2 helicase function in the presence of ATP, non-hydrolyzable AMP-PNP, no ATP, or EDTA. **g** Quantification of ssDNA product from (**f**) ($n = 3$–$10$ biological replicates). **h** DNA melting assay using either wild-type (WT) or K154A ERCC6L2 with the 3' overhang substrate assessing activity in the presence of ATP or non-hydrolyzable AMP-PNP. Assay performed with 60 mM NaCl instead of 20 mM NaCl ($n = 3$–$4$ biological replicates). **i** Quantification of ssDNA product at 5 nM ERCC6L2 from experiments such as in (**h**). **j** Model for the role of ERCC6L2 in staggered DSB repair. Bar graphs in (**c, e, g, i**) correspond to mean ± SD. Statistical significance was determined with a two-tailed Welch's t-test in (**e, g, i**). Source data are provided as a Source Data file.

## Recombinant AsCas12a-Ultra protein production

*Acidaminococcus sp.* Cas12a M537R/F870L (AsCas12a-Ultra)[64] was cloned into an inducible T7 vector containing a C-terminal NLS and 6x His-tag (Supplementary Table 3). The plasmid was transformed into Rosetta (DE3) cells and grown overnight at 37 °C. The following day, one colony was inoculated into 10 mL of liquid LB containing 50 μg/mL kanamycin and 25 μg/mL chloramphenicol and incubated overnight at 37 °C, shaking. The next morning, 1 L of LB + kanamycin/chloramphenicol was inoculated and grown to an $OD_{600}$ ~ 0.7. The culture was transferred to an 18 °C chamber, incubated for 30 min, and then induced with 0.5 mM IPTG for 18 h. Cells were then pelleted and resuspended in lysis buffer (50 mM Tris-HCl, pH 7.5, 500 mM NaCl, 5% glycerol, 1 mM DTT, 0.1% Triton X-100, 1 mM PMSF). The pellet was sonicated for 8 min (10 s on, 10 s off, 20% duty cycle). The resulting lysate was clarified by centrifugation at $18,000 \times g$ for 30 min at 4 °C. The supernatant was passed over a Ni-NTA resin (Thermo Scientific), washed with 5 column volumes of wash buffer (lysis buffer containing 25 mM imidazole), and eluted in elution buffer (50 mM Tris-HCl, pH 7.5, 500 mM NaCl, 500 mM imidazole, 10% glycerol). The eluent was concentrated using a 100 kDa MWCO Ultra-15 filter unit (Amicon) and further purified over a 16/60 Sephacryl S-300 HR size exclusion column (Cytiva) in SEC buffer (10 mM Tris-HCl, pH 7.4, 300 mM NaCl, 0.1 mM EDTA, 1 mM DTT). Fractions were pooled and concentrated using a 100 kDa MWCO Ultra-15 filter unit (Amicon). The concentrated protein was diluted 2× in SEC buffer containing 80% glycerol to a final concentration of 40% glycerol at 40 μM. Aliquots were flash frozen and stored at −70 °C.

## Cell culture

RPE1 and HEK-293T cells (ATCC) were cultured in Dulbecco's Modified Eagle's Medium (DMEM/F12; Invitrogen) supplemented with 10% fetal bovine serum (FBS; Thermo Fisher Scientific) and 100 units/mL streptomycin and 100 mg/mL penicillin (GIBCO). K-562 cells (ATCC) were cultured in RPMI-1640 GlutaMAX supplemented with 10% FBS and 100 units/mL streptomycin and 100 mg/mL penicillin. Primary bone marrow stromal cells (BMSCs) were obtained from Dr. Kevin Rouault-Pierre laboratory and cultured in alpha MEM (Thermo Fisher, Catalog number 32571093) supplemented with 10% FBS mesenchymal stem cell-qualified (Thermo Fisher, Catalog number 12662029), penicillin (100 units/mL), and streptomycin (100 μg/mL) (Sigma-Aldrich). Primary cells were immortalized by infecting them with lentiviruses expressing SV40 and RasG12V (MOI < 1; a gift from Alasdair Russell from the Genome Editing facility at the CRUK Cambridge Institute). All cell lines were grown under 5% $CO_2$ in a humidified chamber. Mycoplasma tests were routinely performed, and no mycoplasma was detected in any of the cultured cell lines.

## Lentivirus production

Per reaction, 1 μg of genome vector, 800 ng of packaging plasmid (Addgene #8455; courtesy of Bob Weinberg), 200 ng of envelope plasmid (Addgene #8454; courtesy of Bob Weinberg), and 6 μL of polyethyleneimine (PEI) were mixed in 200 μl Opti-MEM and incubated for 10 min at room temperature. Each mixture was then added dropwise to a well in a 6-well plate containing $5 \times 10^5$ plated HEK-293T cells. After incubating for 2 days, the supernatant was collected, passed through a 0.2 μm filter, and used immediately or frozen at −80 °C. Cells were transduced in the presence of 8 μg/mL polybrene.

## Cas9 and Cas12a genome editing

100 pmol of Cas9 or AsCas12a-Ultra was complexed with 120 pmol gRNA for 30 min at 37 °C (1:1.2 ratio) to form RNP. $2 \times 10^5$ cells were pelleted, washed 1× with PBS, and resuspended in 20 μL nucleofector solution (Lonza) + 100 pmol RNP. Cells were then electroporated in 20 μL cuvettes using the following programs/kits in a 4D nucleofector

(Lonza): K-562 (SF/FF-120), RPE1 (P3/EA-104), and BMSC (P3/FF-104). Following electroporation, 80 μL of pre-warmed media was added to each well and left to incubate for 10 min. Cells were gently transferred to 6-well plates and left to incubate for 72 h (or 120 h for BMSCs) before harvesting cell pellets for downstream analysis. For editing experiments using inhibitors, the following small molecule inhibitors were added to the media for 72 h post-electroporation: DNA-PKcs inhibitor (1 μM; AZD7648), Polθ inhibitor (3 μM; PolQi2), ATM inhibitor (10 nM; AZD0156), MRE11 endonuclease inhibitor (50 μM; PFM03).

## Cas9/Cas12a CRISPRi screen

K-562 cells stably expressing dCas9-BFP-KRAB (CRISPRi)[65] were transduced with a lentiviral CRISPRi gRNA library containing ~100,000 elements (5 gRNAs per gene + non-targeting controls); Weissman v2[66] at a target multiplicity of infection of 0.3 in the presence of 8 μg/mL polybrene for a target coverage of >500 cells per element. The lentivirus genome contained a CasCherry reporter gene that was used to monitor transduction and also harbored a Cas12a and Cas9 target site in close proximity[67]. Two days post-transduction, cells were split into two replicates and treated with 10 μg/mL puromycin for 7 days. Cells were passaged and fresh puromycin added every 2 days. Nine days post-transduction, cells were electroporated with either Cas9 RNP, Cas12a RNP, or a mock plasmid control. 100 pmol RNP was used per $1 \times 10^6$ cells. After roughly 10 doublings (13 days), cell pellets were collected and flash frozen. Genomic DNA was purified using the Puregene kit (Gentra), and the concentration was measured using a Nanodrop. The gRNA cassette was amplified with NEBNext Ultra II Q5 HiFi polymerase (New England Biolabs) with indexed primers using the following steps with 5 μg of gDNA input per 50 μL reaction: 98 °C for 30 s; 22 cycles of 98 °C for 10 s, 65 °C for 75 s; follow by a final incubation at 65 °C for 5 min. All common reactions were pooled and purified in a double SPRI bead purification. In the first round, 0.65X SPRI beads were added, and the supernatant collected. In the second round, 1X SPRI beads were used, and the amplicon was eluted in 20 μL $H_2O$. The purified amplicons were analyzed on a TapeStation D-1000 HS flow cell (Agilent) for the expected amplicon size of 270 bp. Indexed amplicons were sequenced on a NextSeq 2000 P3 flow cell (Illumina) with 50 bp single end reads with a target of 50 M reads per sample. Reads were demultiplexed and analyzed by DrugZ[68] first at the gRNA level then the gene level to identify enriched and depleted genes. Gene-level ranks comparing Cas12a to Cas9 are listed in Supplementary Data 1. Statistical tests are performed using the Benjamini-Hochberg procedure contained within DrugZ.

## Competition assay

Cells were mixed at a 50:50 ratio by cell number and correct ratios were confirmed by flow cytometry using an Attune NxT Flow Cytometer (Invitrogen). For competition assays involving RNP nucleofection, cells were electroporated as described above. For drug treatments, cells were directly resuspended in the drug-containing cell culture medium. ATM inhibitor (AZD0156) was used at a final concentration of 10 nM.

## Quantitative reverse transcription (RT)-qPCR

Per condition, 200,000 cells were collected washed once in PBS and then total RNA was isolated using the RNeasy Kit (QIAGEN), according to the manufacturer's instructions. Reverse transcription was done using iScript™ Reverse Transcription Supermix (Bio-Rad) and the SsoAdvanced Universal SYBR Green Supermix (Bio-Rad) was used for all reactions. Ten nanogram of input cDNA and 1 μM concentrations of primers was used (Supplementary Table 2) and run on a QuantStudio 6 Flex Real-Time PCR system (Applied Biosystems). Final relative mRNA levels were calculated using the Delta Delta Ct (ΔΔCt) method.

## Illumina sequencing

Seventy-two hours post-editing, genomic DNA was collected for analysis using either QuickExtract (Lucigen) or the DNeasy kit (Qiagen) following the recommended manufacturer's protocols. A two-step amplification approach was used to amplify the genomic region of interest and indexed for multiplexing. In PCR1, genomic DNA was amplified with NEBNext Ultra II Q5 (New England Biolabs) with the following cycle conditions: denaturation at 98 °C for 30 s followed by 25 cycles of 98 °C for 10 s, 60 °C for 15 s, and 72 °C for 20 s. 1 μL of PCR1 was used directly in PCR2 containing Illumina indexing primers (New England Biolabs) with the following cycle conditions: denaturation at 98 °C for 30 s followed by 10 cycles of 98 °C for 10 s and 65 °C for 75 s. Amplicons were then purified using 0.8x SPRI beads. Concentrations were measured on a Qubit and further analyzed for amplicon length and sample purity on a TapeStation D-1000 HS DNA flow cell (Agilent). Pooled samples were sequenced either with a MiSeq 2 × 150 paired-end or a NextSeq2000 2 × 150 paired-end flow cell (Illumina) by the Functional Genomics Centre Zurich (FGCZ) in combination with the Genome Engineering and Measurement Lab (GEML) at ETH Zurich. The target average read count per sample was 100-200k reads. Reads were demultiplexed and analyzed using CRISPResso2 (v2.3.2) and Rational Indel Meta-Analysis (RIMA; v2) pipelines[69].

## Nanopore sequencing

The following steps were carried out in an amplicon-free pre-PCR area. Genomic DNA was collected for analysis using the DNeasy kit (Qiagen) following the recommended protocol with the addition of RNaseA (2 μg/mL) at the proteinase K/lysis step. Genomic DNA was quantified using Qubit Broad-Range dsDNA assay. Five hundred nanograms of DNA were amplified using NEBNext Ultra II Q5 HiFi polymerase (New England Biolabs) with primers containing stubbers for downstream indexing (Supplementary Table 2). Target amplicon lengths were between 3.5 and 4.5 kb surrounding the cut site. The following PCR cycle conditions were used: denaturation at 98 °C for 30 s followed by 25 cycles of 98 °C for 10 s, 60 °C for 30 s, and 72 °C for 5 min. PCR products were purified with 0.8X SPRI beads and eluted in H$_2$O. Libraries were indexed and generated using the PCR Barcoding Expansion 1-96 [EXP-PBC096] for Ligation Sequencing Kit [SQK-LSK114] (Oxford Nanopore). Purified libraries were sequenced on a PromethION with the R10.4.1 flow cell.

Quantification of read lengths was performed using SummarizeOntDels[29]. Briefly, reads were filtered in three steps: (1) Reads containing the PCR primer, (2) Reads containing the first 30 bp of the expected amplicon (excluding primer sequence), and (3) Reads under the maximum allowed length. The filtered reads were then aligned to the human genome (hg38) using minimap2. Read lengths were extracted (bioawk, version 1.0) and counted. For each read length count, the fraction of total reads of greater length was calculated and plotted.

## Cell line generation

For generating RPE1 $TP53^{-/-}$; $ERCC6L2^{-/-}$ clone 47, RPE1 $TP53^{-/-}$; $ERCC6L2^{-/-}$ clone 69.2, RPE1 $TP53^{-/-}$ $NBN^{-/-}$ and $TP53^{-/-}$ $ATM^{-/-}$ 0.6 × 10$^6$ RPE1 $TP53^{-/-}$ cells expressing Cas9 were seeded in 60 mm plates. Cells were transfected with a gRNA targeting each gene using RNAiMAX (Invitrogen) according to the manufacturer's protocol. After 48–72 h, single cells were sorted into 96-well plates on a FACSAria III (BD Biosciences) or a MoFlo Astrios (Beckman Coulter). Successfully gene-edited clones were selected on the basis of TIDE analysis (https://tide-calculator.nki.nl)[70].

## Proliferation assays (IncuCyte)

Real-time cell proliferation was monitored using the IncuCyte S3 Live-Cell Analysis System (Sartorius). 500 cells were seeded per well in a 96-

well plate and imaged every 3 h until cells reached confluency (indicated on the graph). Confluency was analysed by IncuCyte S3 software.

## ERCC6L2 K154R endogenous mutation installation

Modeling of ATP-bound ERCC6L2 was generated using AlphaFold3[71]. A gRNA targeting amino acid K154 in $ERCC6L2$ was manually designed (Supplementary Table 1). A single-stranded donor (ssODN) was designed to incorporate a SNP in the gRNA binding sequence as well as the intended K154R mutation. RPE1 $TP53^{-/-}$ cells were electroporated with 100 pmol Cas9 RNP and 100 pmol ssODN as described above. Seventy-two hours later, single cells were seeded in a 96-well plate and left to grow for 14 days. Individual clones were scaled up and genotyped using Illumina sequencing. Droplet digital PCR (ddPCR) was performed to determine the zygosity of the clones (Bio-Rad).

## Complementation of $ERCC6L2$ deficient cells

First, RPE1 $TP53^{-/-}$; $ERCC6L2^{-/-}$ clone 47 cells were grown for >1 week in tetracycline negative FBS. Then, these cells were infected with a lentivirus expressing TET3G repressor. Transductants were selected with G418 (500 μg/mL) for 7 days. The transduced cells were then infected with either wild-type or K154R mutant FLAG-ERCC6L2 (MOI < 1). These cells were selected with puromycin for 3 days before seeding them to perform experiments.

## Droplet digital PCR (ddPCR)

Primers and probes were designed against K154R locus and $KMT2C$ as a control (Supplementary Table 2). One thousand five hundred nanograms of genomic DNA was digested by HindIII for 3 h at 37 °C. The digested DNA was five-fold diluted in a buffer containing 2 ng/μL sheared salmon sperm DNA (Invitrogen). Droplets were generated using the QX200 Droplet Generator (Bio-Rad). Droplets were then subjected to thermal cycling (10 min at 95 °C, then 40 cycles of 30 s at 94 °C and 1 min at 57 °C, followed by 10 min at 98 °C). Measurement and copy ratio calculations were conducted using a QX200 Droplet Reader and QuantaSoft software (v1.7.4.0917, Bio-Rad) according to the manufacturer's instructions.

## Western blot

Cell lysates were collected by pelleting, washing, and resuspending cells in 1× RIPA buffer supplemented with 1× protease inhibitor cocktail (Thermo Fisher) or in modified Laemmli lysis buffer (1% SDS, 50mMTris ph 6.8, 5 mM EDTA). Lysates were incubated on ice for 15 min, pelleted at 15k × $g$ for 15 min, and the soluble fraction collected. Protein concentration was determined using Bradford assay or nanodrop (for Laemmli lysates). Twenty micrograms of input were electrophoresed on a 4–20% SDS-PAGE gel. A wet transfer to a 0.8 μm nitrocellulose membrane was performed at 4 °C. The membrane was blocked in TBS-T + 5% milk for 45 min followed by overnight incubation at 4 °C in TBS-T + 2% BSA containing primary antibody. The following primary antibodies and concentrations were used: anti-FLAG (1:1000; Sigma Aldrich F1804 or 1:500; Sigma Aldrich F7425), anti-ERCC6L2 (1:1000; Sigma Aldrich HPA022422), anti-GAPDH (1:2000; Cell Signaling Technology 97166 or 1:1000; Sigma Aldrich MAB374), anti-MRE11 (1:1000, Novus Biologicals NB100-142), anti-NBS1 (1:1000; GeneTex GTX70224), anti-CtIP (1:500; Thermo Fisher 61141), and anti-β-Actin (1:1000; Abcam ab8226). The next day, membranes were washed in TBS-T and then incubated in TBS-T + 5% milk + 1:10,000 secondary antibody (LI-COR 800CW and/or 680RD). Membranes were again washed in TBS-T. A final wash in PBS was performed prior to imaging on a LI-COR Odyssey using the 800 and 700 nm channels. For MRE11 and CtIP immunoblotting, 5% BSA was used instead of 5% milk. Uncropped Western blots can be found in the Source Data file.

## CAST-seq

Genomic DNA from edited RPE1 $TP53^{-/-}$ (wild-type or $ERCC6L2^{-/-}$) cells was extracted using the DNeasy kit (Qiagen) according to the manufacturer's protocol. Primer sequences used for CAST-seq are listed in Supplementary Table 2. CAST-seq analyses were performed as previously described[28] with minor modifications[72,73]. In brief, 200 ng of genomic DNA was used as input material for each technical replicate. Libraries were prepared using the NEBNext Ultra II FS DNA Library Prep Kit for Illumina (New England Biolabs). Enzymatic fragmentation of the genomic was aimed at an average length of 500–700 bp. CAST-seq libraries were sequenced on a NextSeq 2000 using 150-bp paired-end sequencing. Two technical replicates for each sample were run and analyzed. All sites that were present in one technical replicate and significant in at least one replicate are reported (Supplementary Table 4). For sites under investigation, the spacer sequence of the gRNA was aligned to the most covered regions for each site (±400 bp)[72].

## Megabase deletion and chromosome arm loss reporter assay

K-562 cells with a single copy GFP integrated on chromosome 7[29] were stably transduced with ZIM3-dCas9-BFP CRISPRi lentivirus (Addgene #188775; a kind gift from Marco Jost and Jonathan Weissman) and sorted for BFP+ cells. This pool of cells was then transduced with a non-targeting gRNA (gNT) or one targeting ERCC6L2 (gERCC6L2). Positively transduced cells were selected with puromycin for 7 days. Cells were assessed for GFP loss by flow cytometry 7 days post-editing. The flow cytometry gating strategy is exemplified in the Source Data file.

## DISCOVER-seq (MRE11 ChIP-seq)

$10 \times 10^6$ wild-type or $ERCC6L2^{-/-}$ RPE1 $TP53^{-/-}$ cells were mixed with 300 pmol of Cas12a + gMulti RNP and electroporated in a 100 μL cuvette in a 4D nucleofector (Lonza). Twelve hours post-electroporation, cells were detached, pelleted, and resuspended in room temperature DMEM (without supplements). Cells were crosslinked with 1% formaldehyde (Thermo Fisher) and incubated for 15 min at room temperature. Formaldehyde was quenched with 125 mM glycine for 3 min on ice. Cells were then pelleted at 4 °C, washed twice with ice-cold PBS, and snap frozen in liquid nitrogen. Pellets were stored at −80 °C prior to processing.

MRE11 ChIP-seq was performed as previously described[30]. Briefly, samples were thawed on ice and lysed using ice cold buffers LB1, LB2, and LB3. The isolated DNA was sonicated to obtain ~300 bp chromatin fragments using a Covaris S2 with the following settings: 12 cycles of duty cycle 5%, intensity 5, 200 cycles per burst for 60 s. Ten micrograms of MRE11 antibody (NBP3-25349; Novus Biologicals) per ChIP were prebound to protein A Dynabeads (Invitrogen). Chromatin was immunoprecipitated with antibody-bound beads, rotating overnight at 4 °C. Dynabeads were washed with RIPA buffer and the DNA was eluted by incubating overnight at 65 °C in elution buffer. For the final clean-up, the samples were digested with Proteinase K and RNase A in TE buffer for 1 h at 55 °C. DNA fragments were purified using the MinElute Kit (Qiagen). Sequencing libraries were prepared using NEBNext Ultra II kit (New England Biolabs). Indexed libraries were pooled and sequenced on a NextSeq2000 (Illumina) with $2 \times 150$ paired-end reads and a target depth of 20 M reads per sample. Bowtie2 was used to align the reads, and MRE11 peak calling was performed using BLENDER2 with a threshold of 2 (-t 2) in combination with manual inspection[30]. Called peaks for both samples are detailed in Supplementary Table 5.

## Apoptosis assay

Patient and healthy donor BMSCs were edited with gTRAC or gMulti and re-plated after 3 days to remove apoptotic cells from nucleofection. 3 days later, cells, including the supernatant, were collected and stained with Annexin V-FITC (Biolegend) and DAPI solution (BD Bioscience) before being analyzed by flow cytometry.

## Clonogenic survival assay

For RPE1 cells, between 300 and 600 cells per well were plated in 6-well plates in technical duplicates or triplicates. After 4 or 24 h, cells were continuously treated with the specified concentrations of etoposide. They were then cultured for 7 to 10 days (or until colonies became visible), followed by fixation and staining with ethanol and 0.05% crystal violet. Colony number and area per well were quantified using ImageJ, averaged across technical replicates, and normalized to the number of colonies in untreated conditions. For BMSCs, cells were edited and between 1000 and 2000 cells were plated 3 days later in 6-well plates. Cells were cultured for 8 to 10 days, fixed, and then stained as described above.

## Proximity ligation assay (PLA)

Ten thousand cells/well were seeded in 96-well imaging plates (PerkinElmer). After 24 h, cells were treated for 30 s with 20 μM etoposide, then cells were left to repair for the indicated times. Cells were pre-extracted with CSK buffer 0.5% Triton X-100 on ice for 2 min. Afterwards, cells were fixed with 2% formaldehyde for 30 min. Proximity ligation assay (PLA) was performed using antibodies described below, along with Duolink In Situ PLA Anti-Mouse Minus and Anti-Rabbit Plus probes (Sigma-Aldrich). The Duolink In Situ Detection Reagents Far-Red Kit (Sigma-Aldrich) was used according to the manufacturer's instructions. Then, cells were stained with 1 μg/mL DAPI at room temperature for 5 min, washed with PBS and imaged using an Opera Phenix system. The image analysis was performed using Harmony software. Antibodies: anti-CtIP (1:500; Santa Cruz Biotechnology sc-28324), anti-MRE11 (1:1000: Novus Biologicals NB100-142), anti-phospho H2AX (1:5000; Sigma Aldrich 05-636), and anti-NBS1 (1:1000, GeneTex GTX70224).

## RPA accumulation

Flow cytometry-based detection of ssDNA by measuring accumulation of chromatin-bound RPA in etoposide-treated cells was performed mainly as described in ref. 74. Briefly, $0.8 \times 10^6$ cells were seeded in 60 mm plates one day before the experiment. The cells were treated with 0.1 mM etoposide or mock-treated for 1 h, with 10 μM EdU added for the last 30 min to label replicating cells. Following the treatments, cells were detached by trypsinisation, washed with ice-cold 1x PBS and pre-extracted with 0.2% Triton X-100 in 1x PBS for 10 min on ice. Pre-extracted cells were washed in 1x PBS containing 3% BSA (PBS/BSA) and fixed in 100 μL of BD Cytofix/Cytoperm buffer (BD Biosciences; 554722) for 15 min at room temperature. Primary antibody staining was performed for 1 h at room temperature in 50 μL of PBS/BSA per sample containing anti-RPA2 (Abcam, ab2175) and anti-phospho H2AX (Cell Signaling Technology, 2577) antibodies at 1:200 dilution. After a wash in PBS/BSA, the cells were incubated with the secondary antibodies (Alexa Fluor 594 and 488-conjugated goat-anti-rabbit and goat-anti-mouse; Molecular Probes) at 1:1000 dilution for 1 h in the dark. EdU was labeled in Click-iT reaction with Alexa-Fluor 647 Azide (Thermo Fisher, A10277). The samples were analysed on BD LSR Fortessa flow cytometer followed by analysis using FlowJo software. The data from untreated cells were also used for the analysis of the cell cycle distribution.

## Immunofluorescence

Fifty thousand cells/well were plated in 24-well plates that are adequate for imaging. The next day, cells were treated for 30 min with 10 μM etoposide and 10 μM EdU. The treatment was removed and the cells were then washed with PBS, and let repair for the indicated times. Before fixation, cells were pre-extracted with CSK buffer 0.5% Triton X-100 on ice for 5 min. Afterwards, cells were fixed with 2%

formaldehyde for 30 min. After fixation, cells were blocked in PBS-BSA 5% for 1 h (for IF against NBS1, the blocking was PBS-BSA 5%-0.5% Triton X-100). Then cells were incubated with primary antibodies in PBS-BSA 5% (in the case of IF against NBS1: PBS-1% BSA-0.1% Triton X-100) at 4 °C overnight. The next day, after primary antibody removal, the cells were washed three times in PBS-0.1% Tween (in the case of IF against NBS1, the washes were with PBS-0.1%Triton X-100), followed by incubation for 1 h with secondary antibody in PBS-5% BSA (in the case of IF against NBS1, the secondary antibody was incubated in PBS-1% BSA-0.1% Triton X-100). Secondary antibody was washed three times with PBS-0.1% Tween (in the case of IF against NBS1, the washes were with PBS-0.1% Triton X-100), and then cells were stained with 1 µg/mL DAPI at room temperature for 5 min. DAPI was washed with PBS, and finally, cells were stored in MilliQ-grade water. An Opera Phenix microscope was used to acquire images which were analyzed by using Harmony software. For differentiating S phase cells, we used EdU positive vs negative cells, and for differentiating G1 vs G2 phase cells, we used DAPI intensity and nucleus size. For detection of BrdU foci, $3 \times 10^4$ cells/well were plated. The day after seeding, cells were treated with 30 µM BrdU for 24 h followed by 10 µM etoposide and 10 µM EdU treatment for 30 min and let repair for 3 h. Antibodies: RPA32 (1:250; Abcam ab2175), anti-phospho H2AX (1:1000; Cell Signaling Technology 2577), and anti-BrdU (1:300; Cytiva RPN20AB).

### Single molecule analysis of resection tracks (SMART) assay

Cells were treated with 10 µM IdU for 24 h, then with 10 µM etoposide for 30 min before either being harvested immediately or allowed to recover in normal medium for 4 h. IdU-labeled and unlabeled cells were resuspended in ice-cold PBS at $2.5 \times 10^5$ cells/mL and mixed at a 1:5 ratio. A 5 µL aliquot of the cell mixture was lysed in 15 µL lysis buffer (200 mM Tris-HCl pH 7.4, 50 mM EDTA, 0.5% SDS) on SuperFrost Plus slides (Thermo Fisher) for 10 min. Slides were then tilted to ~40°, allowing the buffer to flow slowly down the slide, air-dried for 30 min, and fixed in methanol/acetic acid (3:1) at −20 °C for 15 min. After two PBS washes, slides were incubated in 70% ethanol overnight at 4 °C. Slides were then washed twice in PBS, blocked in 3% BSA/PBS for 1 h, and stained with mouse anti-BrdU antibody (1:100; BD Biosciences clone B44) for 2.5 h at room temperature. After three washes in PBS, slides were incubated with Alexa Fluor 488-conjugated anti-mouse secondary antibody (1:100) for 1.5 h at room temperature. Slides were washed three times for 5 min in PBS and coverslips were mounted with ProLong Diamond Antifade Mountant. Images were acquired using a Zeiss Apotome 3 microscope with a 63× oil immersion objective. Representative images were acquired on a Nikon spinning-disk confocal microscope equipped with a Yokogawa CSU-W1 SoRa scanner unit. Resection tract lengths were measured using Fiji/ImageJ v2.9.0.

### Measurement of DNA end resection by qPCR

HA-ER-AsiSI expressing U2OS[38] were infected or not with a virus expressing Cas9 and a guide RNA against ERCC6L2 (MOI > 1). After selecting transductants with 1 µg/mL puromycin for 3 days, cells were expanded and treated with 300 nM 4-hydroxytamoxifen (4-OHT) for 4 h. The cells were trypsinised for collection, and the pellet was used for genomic purification by PureLink Genomic DNA Mini Kit (Invitrogen) according to the manufacturer's instructions. Two micrograms of DNA/sample was either digested with HindIII or BanI (New England Biolabs) or not digested at 37 °C overnight in 50 µL. Samples were heat-inactivated and purified by AMPure XP beads (Beckman Coulter). 25 ng of the samples were used for qPCR reactions to measure single-stranded DNA at loci 200 nt, 950 nt and 1600 nt away from the Chr22 DSB site (Supplementary Table 2). For every sample, ΔCt was determined by subtracting the non-digested Ct value from the corresponding digested Ct value. The proportion of single-stranded DNA formed by resection was calculated by using the following formula ssDNA% = $1/(2^{(\Delta Ct-1)} + 0.5) \times 100$[37].

### CRISPR knockout etoposide screen

RPE1 *TP53*[−/−]; *ERCC6L2*[−/−] cells expressing Cas9 were transduced with the Gattinara library[75] at a multiplicity of infection (MOI) of 0.2, ensuring 500-fold coverage. Transduced cells were then selected with puromycin for eight days and either left untreated or exposed to 45 nM etoposide (IC20) for 18 days. The extraction of genomic DNA was performed by using QIAamp Blood Maxi Kit (Qiagen). Then we used the Q5 Master Mix (New England Biolabs Ultra II) to perform a PCR to amplify gRNA sequences with i7 barcoded multiplexing primers. The purified PCR products were sequenced on an Illumina NovaSeq 6000 system. Guide RNA enrichment analysis was performed using DrugZ to compare DMSO-treated and etoposide-treated samples[68] (Supplementary Data 2). Statistical tests are performed using the Benjamini-Hochberg procedure contained within DrugZ.

### siRNA transfection

$1 \times 10^5$ cells RPE1 *TP53*[−/−] and *TP53*[−/−]; *ERCC6L2*[−/−] cells were first transfected in suspension in 6-well plates. Transfections were performed using 20 nM siRNAs (CtIP: 5′ gcuaaaacaggaacgaauc 3′; Luciferase: 5′ cguacgcggaauacuucga 3′), MRE11: siGENOME Human MRE11 (4361) siRNA−SMARTpool (Horizon). Transfections were carried out using Lipofectamine RNAiMAX (Thermo Fisher Scientific) according to the manufacturer's protocol. After 24 h, the medium was changed and another round of transfection was carried out with attached cells. After 24 h, the medium was changed. 72 h after the first transfection, cells were detached and seeded for clonogenics assays, while the rest were used in immunoblotting for validating knockdown efficiency.

### ERCC6L2 purification

The human ERCC6L2 sequence, codon optimized for the expression in *Sf*9 cells, was purchased from Twist Bioscience (Supplementary Table 2) and cloned into pFB-2XMBP-BRCA1co-his[76] using NheI and XhoI and standard procedures to generate pFB_2xMBP-ERCC6L2co_his (Supplementary Table 3). The K154A mutation was generated using QuikChange XL II kit (Agilent). The protein was expressed in *Spodoptera frugiperda* cells (ExpiSF9™ cells, ThermoFisher Scientific); bacmids, primary and secondary viruses were prepared according to manufacturers' recommendations. The cells were harvested 52 h after infection, washed with phosphate buffered saline, snap frozen in liquid nitrogen, and stored at −80 °C. All subsequent purification steps were carried out at 0–4 °C. The pellets were thawed and resuspended in lysis buffer (Tris-HCl pH 7.5, 50 mM; dithiothreitol, 1 mM; EDTA, 1 mM; Sigma protease inhibitory cocktail, P8340, 1:400; phenylmethylsulfonyl fluoride, 1 mM; leupeptin, 30 µg/mL) and incubated at 4 °C for 20 min. Glycerol was added to a final concentration of 25%, NaCl was added to a final concentration of 325 mM and the cell suspension was incubated at 4 °C for 30 min. The cell suspension was centrifuged at 55,000 × *g* at 4 °C for 30 min to obtain soluble extract. Soluble extract was incubated with amylose resin (New England Biolabs) for 1 h with gentle agitation. The resin was washed with buffer W (Tris-HCl pH 7.5, 50 mM; β-mercaptoethanol, 5 mM; NaCl, 1 M; phenylmethylsulfonyl fluoride, 1 mM, glycerol, 10%) first batchwise, and then extensively on a disposable column by gravity flow (Thermo Scientific). 2xMBP-ERCC6L2-his was eluted with elution buffer (Tris-HCl pH 7.5, 50 mM; β-mercaptoethanol, 5 mM; NaCl, 0.3 M; phenylmethylsulfonyl fluoride, 1 mM; glycerol, 10%; maltose, 10 mM). The eluate was treated with PreScission protease (12.5 mg of protease per 100 mg protein), and incubated for 90 min at 4 °C. Next, imidazole was added to the sample (10 mM final concentration), and the solution was immediately loaded onto pre-equilibrated Ni-NTA agarose resin (Qiagen) at 4 °C, in flow. The resin was washed on column first with Ni-NTA buffer A1 (Tris-HCl pH 7.5, 50 mM; β-mercaptoethanol, 5 mM; NaCl, 1 M; glycerol, 10%; imidazole, 10 mM) and subsequently with Ni-NTA buffer A2 (Tris-HCl pH 7.5, 50 mM; β-mercaptoethanol, 5 mM; NaCl, 150 mM; glycerol, 10%; imidazole, 10 mM). ERCC6L2-his was eluted with

buffer B (Tris-HCl pH 7.5, 50 mM; β-mercaptoethanol, 5 mM; NaCl, 150 mM; glycerol, 10%; imidazole, 300 mM). Fractions containing the protein were pooled, dialyzed 1 h against 1 L dialysis buffer (Tris-HCl pH 7.5, 50 mM; β-mercaptoethanol, 5 mM; NaCl, 150 mM; glycerol, 10%), aliquoted, frozen in liquid nitrogen and stored at -80 °C. The K154A variant was purified as described above for the wild-type protein.

## Human RPA purification

Human RPA was expressed in *Sf*9 insect cells with the pFB-RPA1, pFB-RPA2, pFB-6xhis-RPA3 vectors and purified by Ni-NTA affinity chromatography using first an HiTrap Blue column (Cytiva) followed by desalting with a HiTrap Desalting column (Cytiva) and ultimately with an HiTrap Q column (Cytiva)[76].

## Preparation of DNA substrates

The sequence of the oligonucleotides used for DNA substrate preparation is listed in Supplementary Table 2. The dsDNA substrate was prepared with the oligonucleotides X12-3 and X12-4C; the dsDNA substrates with 4 nt overhang at the 5′ or 3′ terminus were prepared with the oligonucleotide X12-3 annealed to X12-4C_5′Over or X12-4C_3′Over, respectively. The Y-structure substrate was prepared with the oligonucleotides X12-3 and X12-4NC. The X12-3 oligonucleotide was labeled at the 3′ terminus with (α-$^{32}$P) dCTP (Hartmann Analytic) and Terminal Transferase (New England Biolabs) according to manufacturer's instructions. The substrates were prepared by heating the oligonucleotides at 95 °C and slow gradual cooling to room temperature. Unincorporated nucleotides were removed with Micro Bio Spin P-30 Tris chromatography columns (Bio-Rad).

## Helicase assay

Helicase assays were carried out in 15 mL of final volume in buffer containing 25 mM Tris acetate (pH 7.5), 20 mM NaCl, 2 mM magnesium acetate, 1 mM dithiothreitol, 0.1 mg/mL recombinant albumin (New England Biolabs) and 1 nM DNA substrate. Experiments where wild type ERCC6L2 was compared to the K154A mutant were carried out in the same buffer but with 60 mM NaCl. Where indicated, the buffer was supplemented with 1 mM ATP or AMP-PNP (Adenylyl-imidodiphosphate) and/or 15 nM of human RPA. The reactions were assembled on ice and incubated for 30 min at 37 °C. The reactions were terminated with 5 μL of stop buffer (150 mM EDTA, 0.2% SDS, 30% glycerol, 0.1% bromophenol blue) and 1 μL of Proteinase K (14-22 mg/mL, Roche) for 15 min at 37 °C and analyzed by electrophoresis on 10% native polyacrylamide gel in TBE buffer. Gels were dried on CHR-17 paper (Whatman) and detected by autoradiography.

## Electrophoretic mobility shift assay (EMSA)

EMSA assays were carried out in 15 μL of buffer containing 25 mM Tris Acetate (pH 7.5), 20 mM NaCl, 1 mM dithiothreitol, 0.1 mg/mL recombinant albumin (New England Biolabs), 3 mM EDTA, and 1 nM DNA substrate. Experiments where wild-type ERCC6L2 was compared to the K154A mutant were carried out in the same buffer but with 60 mM NaCl. The reactions were assembled and incubated for 15 min on ice. The complexes were then mixed with 5 μL of loading buffer (50% glycerol, 0.1% bromophenol blue) and loaded immediately on 6% polyacrylamide gel in TAE buffer on ice. The gels were dried on CHR-17 paper (Whatman) and analyzed by autoradiography. When competitor DNA was used, the cold oligonucleotides, in the concentrations as indicated, were added to the complexes after the 15 min of incubation on ice. The samples were then incubated for an additional 15 min at 37 °C, returned on ice for 5 min and then mixed with loading buffer and analyzed as above.

## Reporting summary

Further information on research design is available in the Nature Portfolio Reporting Summary linked to this article.

## Data availability

Sequencing data for the genomic screens, short-read, and long-read sequencing are available in the Sequence Read Archive (SRA) as Bio-Project with the accession number PRJNA1299151. Aligned ChIP-seq data are available in NCBI's Gene Expression Omnibus with the GEO Series accession number GSE312241. Analyzed datasets for genomic screens, ChIP-seq, and CAST-seq are available in the Supplementary Information and in Supplementary Data 1 and Supplementary Data 2. Source data are provided with this paper.

## Code availability

The following pipelines have used code available on GitHub: DrugZ (https://github.com/hart-lab/drugz), CRISPResso2 (https://github.com/pinellolab/CRISPResso2), RIMA2, SummarizeOntDels (https://github.com/cornlab/summarizeOntDeletions), first version of CAST-seq analysis (https://github.com/AG-Boerries/CAST-Seq), DISCOVER-seq (MRE11 ChIP-seq) https://github.com/cornlab/blender. Chromosomal representation of gMulti target sites was generated using karyoploteR (version 1.28.0)[77]. Genomic annotations for Cas12a gMulti were retrieved using the annotatr package (version 1.28.0)[78].

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

## Acknowledgements

We would like to thank Rolf Turk at IDT for generously supplying AsCas12a-Ultra used in the screen. We would also like to thank Anna Bratus-Neuenschwander at the Functional Genomics Centre Zürich for great assistance with Nanopore sequencing. We would like to thank Tabea Quaderer for assistance in experiments. We would also like to thank Chris Carnie for all the discussions and feedback he provided to this project. We would also like to thank Alasdair Russell and the whole Genome Editing facility at the CRUK Cambridge Institute, and Huw Naylor, Heather Zecchini and the whole Microscopy Facility at the CRUK Cambridge Institute for all their support. We thank Kate Dry for editorial assistance. We thank Stefan Braunshier (Cejka lab) for human RPA. J.E.C. and S.P.J. are funded by the European Research Council (ERC) under the European Union's Horizon 2020 Research and Innovation Programme (no. 855741-DDREAMM-ERC-2019-SyG). J.E.C. is supported by the NOMIS Foundation and the Lotte und Adolf Hotz-Sprenger Stiftung, Swiss State Secretariat for Education, Research and Innovation (SERI) and SNSF Project Funding grants (310030_188858; 320030_227979). E.J.A. has received support by an EMBO Postdoctoral Fellowship (ALTF 144-2021). J.F. is a recipient of the EMBO Postdoctoral Fellowship (ALTF 220-2021). Research in the SPJ laboratory is supported by Cancer Research UK (CRUK) Discovery Award DRCPGM\100005, CRUK Cambridge Institute core grant SEBINT-2024/100003 and ERC Synergy Award 855741. A.S.B., N.G. and V.G. were supported by ERC Synergy Award 855741; R.B. by CRUK Discovery Award DRCPGM\100005 and a GlaxoSmithKline award to S.P.J.; and A.S.B. by CRUK RadNET Cambridge C17918/A28870 and Wellcome Early Career Award 227014/Z/23/Z. The SPJ laboratory was also supported by core funding grants C6946/A24843 and WT203144 to the Gurdon Institute. P.C. is funded by the Swiss National Science Foundation (SNSF) (Grants 310030_207588 and 310030_205199) and the European Research Council (ERC) (Grant 101018257). T.Ca. was supported by the German Research Foundation (CA 311/4-1 & FANEDIT/EJPRD20-209). G.A. is funded from the German Federal Ministry of Education and Research (BMBF) within the Medical Informatics Funding Scheme EkoEstMed–FKZ 01ZZ2015.

## Author contributions

E.J.A., A.S.B., S.M.S., E.C., R.B., N.G., J.F., and G.C. designed, performed, and analyzed cell-based experiments. A.S.B. and V.G. analyzed cell-based experiments. E.C. and P.C. performed in vitro reconstitution experiments. S.A., G.A., and R.R. performed CAST-seq experiments. M.D.R.G. provided primary BMSCs. T.Ca., P.C., J.E.C., and S.P.J. provided experimental guidance and material support. E.J.A., A.S.B., S.M.S., J.E.C., and S.P.J. wrote the manuscript with contributions from all other authors.

## Competing interests

T.Ca. and G.A. are inventors of CAST-Seq patents (e.g., US11319580B2 and EP3856928B1). J.E.C. is a co-founder and SAB member of Serac Biosciences and an SAB member of Relation Therapeutics, Hornet Bio, and Kano Therapeutics. E.J.A. was partly supported by funding from CSL Behring. S.P.J. works part-time at Insmed Innovation UK Ltd. S.P.J. is a founding partner of Ahren Innovation Capital LLP, a co-founder of Mission Therapeutics Ltd, and is a consultant and shareholder of Genome Therapeutics Ltd. The remaining authors declare no competing interests.
