## [Transparent Peer Review File · Nature Communications]

ERCC6L2 ensures repair fidelity for staggered-end DNA double-strand breaks

Corresponding Author: Professor Stephen Jackson

Version 0:

Reviewer comments:

Reviewer #1

(Remarks to the Author)

The study by Aird et al. was motivated by the goal of identifying new factors that selectively repair either blunted or staggered DNA break ends. Using drop-out CRISPRi screening in the K-562 cell line, the authors identified both known and novel factors involved in the selective repair of blunted DNA double-strand breaks induced by Cas9 and staggered double-strand breaks induced by AsCas12a. Among the newly identified factors, they focused on ERCC6L2, a SWI/SNF family ATPase previously implicated in NHEJ and linked to bone marrow failure and leukemia, to elucidate its role in staggered end-specific DNA repair.

Through a co-culture experiments, they demonstrated that loss of ERCC6L2 did not affect cell fitness. By combining second- and third-generation sequencing approaches, they found that ERCC6L2 deficiency specifically increased the frequency of large deletions (11 bp to 4 kb) in cell line-based assays following staggered-end induction, whereas Cas9-induced deletions occurred independently of ERCC6L2. These observations were reinforced by deletion analyses in primary human bone marrow-derived mesenchymal stem cells (BMSCs) from healthy donors and ERCC6L2-deficient donors, as well as in ERCC6L2 knockout cell lines with staggered ends introduced via TALENs or dual Cas9 nickases.

The authors further performed a synthetic drop-out CRISPR-Cas screen in RPE1 cells deficient in both ERCC6L2 and p53 under etoposide treatment. NBN (NBS1) and ATM were found to be the top candidates. Co-culture experiments showed that loss of NBS1 protected ERCC6L2-deficient cells from etoposide-induced cytotoxicity. Flow cytometry revealed that ERCC6L2-deficient cells exhibited an increased proportion of G2-phase cells double-positive for RPA and γ H2AX, suggesting that ERCC6L2 suppresses prolonged resection under etoposide treatment. Inhibiting ATM rescued etoposide-induced lethality in ERCC6L2-deficient cells and reduced the number of RPA-positive cells in G2 phase, as well as decreased long-range resection.

Finally, biochemical assays identified ERCC6L2 binding substrates and demonstrated that its substrate-melting activity was partially dependent on lysine 154 within the ATP-binding pocket.

This manuscript presents compelling and comprehensive results for the role of ERCC6L2 in protecting staggered DNA double-strand break ends. The experiments are well-designed, but lack of sufficient repeats in some cases, and logically executed, and the manuscript is presented in an engaging manner. The findings are likely to be of broad interest to the Nature Communications readership. We have several questions for the authors that, when addressed, could strengthen the conclusions of this work.

Below are the reviewer's remarks to the authors.

- The screen design inherently allows for two possible causes of dropout: reduced cell fitness versus deficiency in blunt or staggered end repair. The data in Figure 1b and Extended Data Figures 1e-f are somewhat unclear in this respect. In Figure 1b, direct comparison of Cas9 and Cas12a suggests a subset of factors that drop out in an end-feature-dependent manner. However, in Extended Data Figure 1b, comparison of Cas9 or Cas12a to the mock control shows that several factors, including ERCC6L2, are significantly depleted in both Cas conditions. Comparison to the mock control should control for the confounding effect of cell fitness, which would suggest that ERCC6L2 also contributes to repair of Cas9-mediated DSBs. Could the authors clarify this discrepancy?

- Related to this point, could the absence of NHEJ end protectors or chaperones (Ku70/80 or Artemis/XLF) expose Cas9-generated blunt ends to limited resection, thereby rendering them accessible to ERCC6L2 activity?
- Following prior points, the authors wrote, In lines 82–84, “the authors state that ERCC6L2 “played no role in Cas9-induced DSB repair” but was essential for Cas12a-induced DSB repair.” This conclusion does not align with Extended Fig. 1e findings.
- ARID1A appears among the top differential hits in the initial screen and is known to function in both NHEJ and HR, with high mutation frequency in cancer. Given its potential relevance, could the authors briefly comment on its end-structure specificity in their dataset and the rationale for focusing primarily on ERCC6L2?
- Figures 3a–b: The authors report that 28 Cas12a-induced cuts in ERCC6L2-deficient cells are sufficient to cause lethality. This is a striking result. Has it been validated that only 28 cuts were introduced, or could the sgRNA have generated multiple off-target sites? For example, was this assessed using CAST-seq or other equivalent translocation sequencing approaches?
- Figure 3h: The triangular relationship between RPA32 intensity and resection length is intriguing. Has this correlation been previously reported? If so, could the authors provide an appropriate reference?
- The paper does not have a limitations section in the discussion. General limitations should be discussed. The small number of replicates supporting some of the claims should be included as a limitation.
- Along the line above, across multiple long-read sequencing experiments (e.g., Fig. 1g–h, Fig. 2 and in extended Figs. 1-2, 3), there were only one technical repeat per condition. Given that several major conclusions—such as the differential impact of ERCC6L2 on Cas12a- versus Cas9-induced large deletions—depend on these data, could the authors clarify whether additional independent replicates exist? If not, it would be helpful to acknowledge this limitation explicitly in the Discussion and temper related inferences.
- Lines 903–916: In many cases, only one replicate is made (e.g. bar graphs in Fig. 2). Care should be taken in the text to ensure it is understood that these results are exploratory/illustrative and that broader inferences cannot be made without further replicates and robust statistical testing. The same point applied in several other places eg (462–479, 886–901, 543–549)
- Lines 289-291: “The authors write: Despite ERCC6L2 binding all dsDNA substrates tested, we observed that it melted dsDNA substrates with either 5' or 3' four nucleotide overhangs but not blunt-ended DNA or Y-structured DNA (Fig. 4d,e)” In many cases in the paper, the authors only have one or two technical replicates. This has been commented on already. However, here the authors make inferential claims, but do not carry out any statistical testing between conditions, despite having five biological replicates. The same point applies for Fig 4 f,g (lines 295-298). The authors should carry out statistical testing (eg ANOVA) between these different conditions.
- Following the same experiments, in Fig. 4f–i, the authors conclude that DNA melting by ERCC6L2 is strongly ATP-dependent. However, AMP-PNP retains partial activity, and differences between 5' and 3' overhang substrates are apparent but not discussed. Replicate quantification with statistical analysis would strengthen the claim of ATP dependence and address potential substrate-specific effects.
- Could the authors demonstrate the loss of function results for their genetic modified cell lines?
- In lines 169–172, the TALEN experiments are described as shown in “Extended Fig. 3f,g,” but there does not appear to be an Extended Fig. 3g in the current version.
- For all the co-culture experiments: reciprocal experiments (swapping genotypes for the GFP-positive and negative cells, for example) would exclude cell culture clonal effect.

Reviewer #2

(Remarks to the Author)

Reviewer #3

(Remarks to the Author)

The authors perform a genome-wide CRISPRi screen using a CasCherry reporter cleaved by either Cas9 or Cas12, and identify ERCC6L2 specifically required for the DNA double-strand breaks with a staggered end caused by Cas12. Mechanistically, the author proposed that ERCC6L2 counteracts ATM-driven reaction by binding and melting staggered

DNA ends to promote accurate end joining. This study offers important new insights not only into how cells repair different types of DNA double-strand breaks but also provides a rationale behind the choice of using Cas9 versus Cas12 for future therapy using CRISPR technologies. The study is well-executed, with strong data supporting its conclusion. The weakness of this study lies in the lack of strong data supporting that ERCC6L2 prevents DSB resection specifically at the staggered end. I would strongly support the publication of this study in Nature Communications, provided that the comments below are addressed.

Main comments:

-Is ERCC6L2 recruited to the DNA double-strand break caused by ETP or Cas12?

- NBS1 was found to increase WT cell sensitivity but not ERCC6L2 to ETP. Is a similar phenotype also observed following depletion of MRE11 or RAD50, which are in complex with NBS1? Based on their interpretation that this phenotype reflects a resection defect, one would expect this to be the case. Yet MRE11 and RAD50 were not highlighted in the CRISPR screen results shown in Figure 3e. Similar comment for CTIP depletion.

- Because the difference in RPA staining between WT versus ERCC6L2 is very subtle, especially in S-phase, where most of the resection normally occurs (see Figure 3g), additional information supporting that ERCC6L2 prevents hyperresection of the DSB would strengthen the manuscript. Does depletion of ERCC6L2 enhance ATR signaling that is activated after resection of the break by monitoring RPA or Chk1 phosphorylation? Do ERCC6L2 KO cells show increased levels of ssDNA at the break after ETP using BrdU detection (PMID: 34888531)? Can the author confirm that they don't see such changes after DNA damage-inducing DSB with blunt ends?

- The interpretation of the results from the cell competition and clonogenic survival assays may be misleading, as they could reflect a severe growth defect rather than increased cell death. Can the authors demonstrate elevated cell death in ERCC6L2-deficient cells compared to WT using a cell death marker such as Annexin V or Caspase cleavage?

Other comments:

- There is a lack of statistical analysis across the manuscript. Moreover, if the analysis of the >1 kb deletions using sequencing data was obtained from a single replicate (as there is no error bar in these graphs), this should be indicated in the figure legend.

- In Figure 6d, the number of NSB1 foci/nucleus should be shown for each individual cell measured.

- In addition, individual data points should also be indicated for Figures 3g, 3j, S5f, and S6g.

- The legends at the top of the graph in Figure 3f and 3i should be made consistent (missing NBS-/- in 3f)

- There is a missing data point in Figure 3d (ERCC6L2 at 300nM ETP). Additionally, it would be beneficial to provide an example of clonogenic survival.

- It is unclear how the ERCC6L2 KO clones were validated by RT-qPCR. Is the author measuring mRNA expression (mRNA levels are often not affected by deletion events) or specific deletion events? This should be clearly indicated in the method section.

- While Sanger sequencing was used to validate the NBS1 knockout clones, it remains unclear to what extent NBS1 is depleted in these clones. A Western blot analysis should be included to confirm protein loss, especially since, to my knowledge, reliable NBS1 antibodies are available, or by PCR

Reviewer #4

(Remarks to the Author)

Reviewer #5

(Remarks to the Author)

Aird et al. investigate the differential requirements for repairing DSBs induced by different genome editing strategies, specifically comparing CRISPR-Cas9 and Cas12a, which produce blunt and staggered DSBs respectively. Using a clever dual-PAM reporter construct that enables delivery of both DSB types, combined with a tailored DDR gRNA library, they discovered that ERCC6L2 is preferentially required for protecting staggered DSB ends. The authors generated ERCC6L2 knockout and helicase-defective mutants and demonstrated through both short- and long-read sequencing that these cells accumulate large deletions and translocations. This protective role is independent of the nuclease generating staggered DSBs, as evidenced by similar phenotypes with Cas12a, TALEN-FokI, and dual Cas9 nickases. Through synthetic lethality screening with and without etoposide treatment, they found that while ERCC6L2-deficient cells are hypersensitive to etoposide, NBS1 (MRN complex member) depletion causes hypersensitivity in wild-type cells but does not further sensitize

ERCC6L2-deficient cells, suggesting these factors operate in the same pathway. The authors demonstrate that ERCC6L2 limits ATM-dependent resection, as ATM inhibition reduces both resection and ERCC6L2-deficient cell sensitivity to staggered DSBs, while decreasing large deletion formation. Finally, biochemical analysis of purified ERCC6L2 protein reveals preferential binding to dsDNA and specific melting activity toward dsDNA substrates containing four-nucleotide overhangs, independent of overhang polarity (5' or 3').

The study is well-conducted and the data presented in this manuscript support the conclusion that ERCC6L2 prevents staggered DNA ends from forming large deletions and translocations. This is particularly relevant in etoposide-treated cells, which leave a 4-nucleotide overhang. My major experimental concerns involve the lack of complementation experiments with the clones and the use of an indirect measure of end resection, which is at the core of the proposed mechanism. Conceptually, I think the article lacks analysis of which pathway takes over in the ERCC6L2-deficient context, which has clinical relevance and could be easily addressed, and the fact that a more extensive discussion is needed of how the end-melting activity of ERCC6L2 would actually impede end-resection and the temporal relationship with ATM. Having said that, if the authors address these major concerns, I think readers of Nature Communications will enjoy this manuscript.

List of Major concerns:

1. In Figure 2, the authors generated two independent knockout clones but fail to provide complementation experiments in the analysis of deletion generation through sequencing. The authors should provide a complementation experiment with the full length protein.
2. It is unclear which repair pathway takes over in ERCC6L2^{-/-} cells, which could have relevance for patients. Since POLQ appeared in their screen, do the authors think MMEJ is responsible for repairing breaks not protected by ERCC6L2? The authors should analyze the sequencing experiments for evidence of microhomology in the larger deletions and inhibit POLQ and test viability in the presence of staggered DSBs versus blunt DSBs. It would be interesting to test whether the translocations observed in BMSCs are MMEJ-dependent. This information is potentially important for patients with biallelic mutations in ERCC6L2.
3. The authors propose that ERCC6L2 protects against resection mediated by CtIP. However, the only experimental setup in which they test this directly is RPA intensity after short and chronic etoposide treatment, which is indirect. The authors should provide another way to measure end-resection at breaks, for example by performing qPCR around the TRAC gRNA site to measure the extent of resection. If this proves difficult due to Cas12a-induced ssDNA degradation (Chen et al., 2018) that could confound the results, the authors could use the AsiSI enzyme to induce breaks and measure resection at extensively validated AsiSI-induced breaks.
4. While the authors propose that ERCC6L2 prevents end-resection by melting DNA ends, how this melting prevents end-resection while favoring NHEJ is not entirely clear. How ERCC6L2 is recruited to breaks and what the temporal relationship is between ERCC6L2 end melting and ATM-dependent resection remains unclear. Does ERCC6L2 arrive before MRN, perform the melting, and thereby impede ATM activation and subsequent end-resection? The authors should at least discuss this in the discussion.
5. Throughout the article, consideration of the cell cycle is missing. Do ERCC6L2-deficient cells show an impaired cell cycle that changes the repair pathway? Is ERCC6L2 activity regulated during the cell cycle? Is this melting activity present only during S-phase, or do the authors think it might also work in G1? This should at least be taken into consideration during the discussion.

Minor comments:

1. In figure 1f would it be worth to include also the 0 bp events, that account for both perfect NHEJ or unedited events? Having the 0bp events will make it easier for the reader to read the graphs normalized by the cutting efficiency.
2. In Figure 3, the authors found NBS1, one subunit of the MRN complex, but not MRE11 or RAD50 or CtIP. While CtIP is commonly essential in DepMap and MRE11 is probably essential in RPE1, RAD50 should not be. Why were they not discovered in the screen? Can the authors perform another validation curve as in panel e, inhibiting both ERCC6L2 and at least one other MRN complex subunit to verify rescue of etoposide sensitivity?
3. In fig 3a the author showed that multiple DSBs induce growth defects in ERCC6L2^{-/-}, contrary to a single DSB induced in Fig. 1d. The author suggested that the difference in viability outcome is due to the fact that multiple DSBs induce large genomic aberrations, while a single DSB only induce large deletions. But in figure 2j the author measured translocation in ERCC6L2 KO with a single DSB. The author should sustain this statement by analyzing chromosomal aberrations on a metaphase spreads in both condition (1 gRNA vs gMulti), (or any other experimental set up they like) to test this hypothesis.
4. Fig3.d some error bars are missing, and one is black other than grey.
5. Fig. Ex2a the TIDE analysis on the left panel is barely visible, increase the size or remake the graph.
6. In Figure 3d, the authors measure growth of ERCC6L2 knockouts with or without etoposide in p53-positive or p53-negative cells. Do the knockouts grow similarly to wild-type cells in p53⁺ and p53⁻ contexts? It would be helpful to provide growth curves showing the difference in Extended Figure 5, along with cell cycle analysis to determine whether ERCC6L2 knockout impairs cell cycle progression and therefore repair pathway utilization.
7. In Figure 6d, the authors used a short but very high dose of etoposide to monitor NBS1 foci formation. On average, they observed 2-3 foci per nucleus. While I agree with the interpretation that there seems to be no difference in repair of those breaks with or without ERCC6L2, the large error bars and the fact that breaks are not fully repaired suggest these conditions are not ideal for testing this. How many breaks did the authors expect with this treatment? Can the authors provide γ H2AX data under the same conditions? Can they use gMulti to induce multiple breaks that should give a higher number of NBS1 foci? Or how many phospho-RPA foci did they observe at 3 hours? While not identical conditions, this would give readers an idea about the expected number of DSBs.

Version 1:

Reviewer comments:

Reviewer #1

(Remarks to the Author)

The authors have addressed the reviewers' concerns regarding reproducibility, statistical testing, the limited role of ERCC6L2 in NHEJ, and have added an analysis of the role of ERCC6L2 in the absence of NHEJ. The authors have also corrected other minor errors. The reviewer believes that this manuscript is now ready for publication in Nature Communications.

Reviewer #2

(Remarks to the Author)

Reviewer #3

(Remarks to the Author)

The authors have addressed all of my comments thoroughly. I strongly support the publication of this manuscript in Nature Communications.

Minor comments:

Why is there such an extensive BrdU signal in the SMART assay for untreated samples, which should not exhibit double-strand breaks leading to DNA end resection?

Reviewer #4

(Remarks to the Author)

Reviewer #5

(Remarks to the Author)

The authors addressed all my comments with solid new experiments and an improved discussion. The paper is now suitable for publication.

made.

Response to reviewers' comments

We thank our reviewers for taking time to evaluate our manuscript. Here we provide a detailed point-by-point rebuttal to all points raised by the reviewers, including data to address their key concerns. Please find below the reviewers' comments in black, with our responses in blue.

Reviewer #1 (Remarks to the Author)

The study by Aird et al. was motivated by the goal of identifying new factors that selectively repair either blunted or staggered DNA break ends. Using drop-out CRISPRi screening in the K-562 cell line, the authors identified both known and novel factors involved in the selective repair of blunted DNA double-strand breaks induced by Cas9 and staggered double-strand breaks induced by AsCas12a. Among the newly identified factors, they focused on ERCC6L2, a SWI/SNF family ATPase previously implicated in NHEJ and linked to bone marrow failure and leukemia, to elucidate its role in staggered end-specific DNA repair.

Through a co-culture experiments, they demonstrated that loss of ERCC6L2 did not affect cell fitness. By combining second- and third-generation sequencing approaches, they found that ERCC6L2 deficiency specifically increased the frequency of large deletions (11 bp to 4 kb) in cell line-based assays following staggered-end induction, whereas Cas9-induced deletions occurred independently of ERCC6L2. These observations were reinforced by deletion analyses in primary human bone marrow-derived mesenchymal stem cells (BMSCs) from healthy donors and ERCC6L2-deficient donors, as well as in ERCC6L2 knockout cell lines with staggered ends introduced via TALENs or dual Cas9 nickases.

The authors further performed a synthetic drop-out CRISPR-Cas screen in RPE1 cells deficient in both ERCC6L2 and p53 under etoposide treatment. NBN (NBS1) and ATM were found to be the top candidates. Co-culture experiments showed that loss of NBS1 protected ERCC6L2-deficient cells from etoposide-induced cytotoxicity. Flow cytometry revealed that ERCC6L2-deficient cells exhibited an increased proportion of G2-phase cells double-positive for RPA and γ H2AX, suggesting that ERCC6L2 suppresses prolonged resection under etoposide treatment. Inhibiting ATM rescued etoposide-induced lethality in ERCC6L2-deficient cells and reduced the number of RPA-positive cells in G2 phase, as well as decreased long-range resection.

Finally, biochemical assays identified ERCC6L2 binding substrates and demonstrated that its substrate-melting activity was partially dependent on lysine 154 within the ATP-binding pocket.

This manuscript presents compelling and comprehensive results for the role of ERCC6L2 in protecting staggered DNA double-strand break ends. The experiments are well-designed, but lack of sufficient repeats in some cases, and logically executed, and the manuscript is presented in an engaging manner. The findings are likely to be of broad interest to the Nature Communications readership. We have several questions for the authors that, when addressed, could strengthen the conclusions of this work.

Below are the reviewer's remarks to the authors.

Comment 1.1

The screen design inherently allows for two possible causes of dropout: reduced cell fitness versus deficiency in blunt or staggered end repair. The data in Figure 1b and Extended Data Figures 1e–f are somewhat unclear in this respect. In Figure 1b, direct comparison of Cas9 and Cas12a suggests a subset of factors that drop out in an end-feature–dependent manner. However, in Extended Data Figure 1b, comparison of Cas9 or Cas12a to the mock control shows that several factors, including ERCC6L2, are significantly depleted in both Cas conditions. Comparison to the mock control should control for the confounding effect of cell fitness, which would suggest that ERCC6L2 also contributes to repair of Cas9-mediated DSBs. Could the authors clarify this discrepancy?

We thank this reviewer for pointing this out. A Cas9 DSB (**Supplementary Fig. 1e**) has a relatively mild effect on the abundance of *ERCC6L2* gRNAs in the screen. Cas9 predominately produces blunt DSBs but can also produce a sub-population of 1 bp overhang DSBs in a sequence-dependent manner (Longo GMC, et al. 2025). We speculate that these 1 bp overhanging DSBs can act as substrates for ERCC6L2 to a certain extent as evidenced by the 10% decrease of 1 bp indels in Cas9-edited cells at *AAVS1* in *ERCC6L2* KO cells compared to WT cells (**Supplementary Fig. 2k**). However, the majority of blunt DSBs from Cas9 are ERCC6L2 independent (**Fig. 2a**, **Supplementary Fig. 2g**), leading to the much larger Cas12a effect size in the screen (**Supplementary Fig. 1f**). We note that ERCC6L2 recently appeared as a modulator of 1 bp indels in a genome-wide screen for factors that influence Cas9 editing outcomes (de Alba EL et al. 2025), though this was not commented on by the authors. We have included mention of these findings in the discussion.

Comment 1.2

Related to this point, could the absence of NHEJ end protectors or chaperones (Ku70/80 or Artemis/XLF) expose Cas9-generated blunt ends to limited resection, thereby rendering them accessible to ERCC6L2 activity?

We used DNA-PK inhibition as a model for the absence of canonical NHEJ factors. Upon DNA-PK inhibition, we observed an increase in large deletions in the absence of ERCC6L2 relative to WT cells in Cas9 edited cells (**Supplementary Fig. 7a-c**). This indeed suggests that when canonical NHEJ is inhibited at blunt breaks, ERCC6L2 becomes necessary to prevent excessive end resection.

Comment 1.3

Following prior points, the authors wrote, In lines 82–84, “the authors state that ERCC6L2 “played no role in Cas9-induced DSB repair” but was essential for Cas12a-induced DSB repair.” This conclusion does not align with Extended Fig. 1e findings.

While ERCC6L2 scores on the depleted side in Cas9 treatment during genome-wide screening (**Supplementary Fig. 1e**), it has comparatively a much smaller effect size than in the Cas12a treated population (**Fig. 1b, Supplementary Fig. 1f**). Through extensive individual experiments in the rest of the manuscript, we found that *ERCC6L2* loss has a phenotype specifically with staggered DSBs. However, the statement in lines 82-84 perhaps anticipates these individual experiments too much and we agree the conclusion in lines 82-84 is too strong given the nuance behind Cas9 blunt versus +1 bp staggered outcomes. We have reworded the statement to instead say “...played a *minimal* role in Cas9-induced DSB repair”.

Comment 1.4

ARID1A appears among the top differential hits in the initial screen and is known to function in both NHEJ and HR, with high mutation frequency in cancer. Given its potential relevance, could the authors briefly comment on its end-structure specificity in their dataset and the rationale for focusing primarily on ERCC6L2?

Unlike ERCC6L2, ARID1A was strongly depleted in both the Cas9 and Cas12a conditions (**Supplementary Fig. 1e,f**), suggesting that it plays an important role in maintaining sequence integrity regardless of end architecture. We focused on ERCC6L2 due to its depletion in both the Cas12a (staggered break) and etoposide (staggered break) screens (**Fig. 1b, Fig. 4d**).

Comment 1.5

Figures 3a–b: The authors report that 28 Cas12a-induced cuts in ERCC6L2-deficient cells are sufficient to cause lethality. This is a striking result. Has it been validated that only 28 cuts were introduced, or could the sgRNA have generated multiple off-target sites? For example, was this assessed using CAST-seq or other equivalent translocation sequencing approaches?

To address these questions, we have now used DISCOVER-seq Cas off-target identification (Wienert B et al 2019) to assess the number of targeted genomic loci. Of the 28 predicted sequence-based gMulti target sites, we observed distinctive MRE11 signal at 25 of them (**Supplementary Fig. 5c,d**). Additionally, two off-target sites were identified, both of which contained a single bp mismatch at the 3' end of the gRNA. There are therefore a total of 27 cuts introduced by Cas12a gMulti.

Comment 1.6

Figure 3h: The triangular relationship between RPA32 intensity and resection length is intriguing. Has this correlation been previously reported? If so, could the authors provide an appropriate reference?

We reasoned that, since RPA is coating ssDNA, longer resection leads to a higher intensity RPA focus. Indeed, it has been published that RPA intensity increases proportionally with the distance travelled by resection nucleases (Soniati MM et al. 2019). To orthogonally measure resection length, we now also performed a SMART assay. SMART relies on a modified DNA combing technique to measure resection progression at the level of individual DNA fibers. We found that following etoposide treatment, *ERCC6L2*^{-/-} cells had a median resection length of 21.45 μm compared to 16.64 μm in wild-type cells (**Fig. 3l and m**).

Comment 1.7

The paper does not have a limitations section in the discussion. General limitations should be discussed. The small number of replicates supporting some of the claims should be included as a limitation.

We have now performed biological replicates for all editing experiments in isogenic clones and patient-derived bone marrow stromal cells (**Fig. 2, Supplementary Fig. 2,3,4,7,8h**). In the discussion, we have now included that the precise roles of DNA melting, end alignment, and resection during ERCC6L2-mediated NHEJ at staggered ends remain unclear.

Comment 1.8

Along the line above, across multiple long-read sequencing experiments (e.g., Fig. 1g–h, Fig. 2 and in extended Figs. 1-2, 3), there were only one technical repeat per condition. Given that several major conclusions—such as the differential impact of ERCC6L2 on Cas12a- versus Cas9-induced large deletions—depend on these data, could the authors clarify whether additional independent replicates exist? If not, it would be helpful to acknowledge this limitation explicitly in the Discussion and temper related inferences.

We have now performed biological replicates for all editing experiments in isogenic clones and patient-derived bone marrow stromal cells (**Fig. 2, Supplementary Fig. 2,3,4,7,8h**). This includes replicates of many long-read sequencing experiments.

Comment 1.9

Lines 903–916: In many cases, only one replicate is made (e.g. bar graphs in Fig. 2). Care should be taken in the text to ensure it is understood that these results are exploratory/illustrative and that broader inferences cannot be made without further replicates and robust statistical testing. The same point applied in several other places eg (462–479, 886–901, 543–549)

We have now performed biological replicates for all editing experiments in isogenic clones and patient-derived bone marrow stromal cells and obtained consistent results (**Fig. 2, Supplementary Fig. 2,3,4,7,8h**).

Comment 1.10

Lines 289-291: “The authors write: Despite ERCC6L2 binding all dsDNA substrates tested, we observed that it melted dsDNA substrates with either 5' or 3' four nucleotide overhangs but not blunt-ended DNA or Y-structured DNA (Fig. 4d,e)” In many cases in the paper, the authors only have one or two technical replicates. This has been commented on already. However, here the authors make inferential claims, but do not carry out any statistical testing between conditions, despite having five biological replicates. The same point applies for Fig 4 f,g (lines 295-298). The authors should carry out statistical testing (eg ANOVA) between these different conditions.

Statistical testing in the form of Welch’s t-test is now included for all bar graphs in new **Fig. 5** (old Fig. 4).

Comment 1.11

Following the same experiments, in Fig. 4f–j, the authors conclude that DNA melting by ERCC6L2 is strongly ATP-dependent. However, AMP-PNP retains partial activity, and differences between 5' and 3' overhang substrates are apparent but not discussed. Replicate quantification with statistical analysis would strengthen the claim of ATP dependence and address potential substrate-specific effects.

Statistical testing in the form of Welch’s t-test is now included for all bar graphs in new **Fig. 5** (old Fig. 4). The reviewer is correct to point out that the observed DNA melting was slightly higher with 3' vs. 5' overhang, but as we do not know whether this is due to a sequence difference or the actual end chemistry, we chose not to elaborate on this point. We also note that the ATPase activity promoted, but was not essential for the observed

DNA melting, which is consistent with what was found previously for the MRN complex (Cannon B et al. 2013, (Paull TT, et al. 1999), (Sharma S et al. 2021).

Comment 1.12

Could the authors demonstrate the loss of function results for their genetic modified cell lines?

We confirmed our *ERCC6L2* KO isogenic clones via genotyping (**Supplementary Fig. 2a, 6b**). Due to the low endogenous expression of *ERCC6L2*, we found that we could not visualize *ERCC6L2* by western blot using any of several antibodies. Instead, we have now performed genetic complementation experiments in the *ERCC6L2* KO cells with both WT and K154R *ERCC6L2* cDNA to determine whether the observed phenotypes are specific to *ERCC6L2* KO. These experiments used inducible FLAG-*ERCC6L2* lentiviral constructs (**Supplementary Fig. 3g, 6d**). Re-expression of WT *ERCC6L2* completely rescued the sequence loss phenotype (**Fig. 2f**) and etoposide hypersensitivity (**Fig. 3i**). Together, these new data demonstrate that the phenotypes observed in the *ERCC6L2* KOs are specific to loss of function of *ERCC6L2*.

Comment 1.13

In lines 169–172, the TALEN experiments are described as shown in “Extended Fig. 3f,g,” but there does not appear to be an Extended Fig. 3g in the current version.

Thank you for noticing this error. The TALEN data are now shown as new **Supplementary Fig. 4a-c**.

Comment 1.14

For all the co-culture experiments: reciprocal experiments (swapping genotypes for the GFP-positive and negative cells, for example) would exclude cell culture clonal effect.

In **Fig 1c,d** we are expressing GFP in cells also expressing sg*ERCC6L2* while the control we are not using any fluorophore. In contrast, in **Fig. 3a** we are expressing mCherry fluorophore in the control and not expressing any fluorophore in cells expressing sg*ERCC6L2*. Furthermore, for the competition assay where WT and *ATM* KO cells were expressing either sg*LacZ* and mCherry or sg*ERCC6L2* and GFP (**Fig. 4a**), we have now swapped the fluorophores: the sgRNA against *LacZ* is in a GFP expressing vector and the sgRNA against *ERCC6L2* (sg*ERCC6L2* #1) is in an RFP expressing vector. We infected WT and *ATM* KO cells with the swapped fluorophores and found a similar trend as we previously observed (**Fig. 4a**). We still observed that *ERCC6L2* deficiency promotes higher hypersensitivity to etoposide in WT than in *ATM* deficient cells, indicating

that the fluorophore doesn't affect the result. We did not include this new data in the revised manuscript to maintain conciseness and clarity.

Reviewer #2 (Remarks to the Author)

Reviewer #3 (Remarks to the Author)

The authors perform a genome-wide CRISPRi screen using a CasCherry reporter cleaved by either Cas9 or Cas12, and identify ERCC6L2 specifically required for the DNA double-strand breaks with a staggered end caused by Cas12. Mechanistically, the author proposed that ERCC6L2 counteracts ATM-driven reaction by binding and melting staggered DNA ends to promote accurate end joining. This study offers important new insights not only into how cells repair different types of DNA double-strand breaks but also provides a rationale behind the choice of using Cas9 versus Cas12 for future therapy using CRISPR technologies. The study is well-executed, with strong data supporting its conclusion. The weakness of this study lies in the lack of strong data supporting that ERCC6L2 prevents DSB resection specifically at the staggered end. I would strongly support the publication of this study in Nature Communications, provided that the comments below are addressed.

Main comments:

Comment 3.1

Is ERCC6L2 recruited to the DNA double-strand break caused by ETP or Cas12?

Recruitment to a Cas12a DSB is difficult to assess, since Cas-fluorophore fusions do not readily form foci at non-repetitive loci (reviewed in Park EJ et al. 2025). Therefore, to address this question, we used a proximity ligation assay (PLA) to quantify the interaction of ERCC6L2 and γ H2AX after 10 μ M etoposide treatment. Since endogenous ERCC6L2 is lowly expressed, we transduced doxycycline-inducible FLAG-ERCC6L2. The PLA signal increases over time, but there is a high background in the non-transduced cells. Although there is a clear trend, we feel that the data are not strong enough to include in the manuscript. However, other studies have shown that ERCC6L2 is recruited to damage sites induced after laser micro-irradiation immediately after Ku and XLF, and before NBS1 (Liu X et al. 2020), (Olivieri et al. 2020). Furthermore, ERCC6L2 interacts with CYREN and Ku70/80 (Liu X et al. 2020).

Comment 3.2

NBS1 was found to increase WT cell sensitivity but not ERCC6L2 to ETP. Is a similar phenotype also observed following depletion of MRE11 or RAD50, which are in complex with NBS1? Based on their interpretation that this phenotype reflects a resection defect, one would expect this to be the case. Yet MRE11 and RAD50 were not highlighted in the CRISPR screen results shown in Figure 3e. Similar comment for CtIP depletion.

To address these questions, we have now used siRNA knockdown of MRE11 and CtIP. We depleted either MRE11 or RBBP8 (CtIP) via siRNA knockdown in *ERCC6L2* KO or WT cells. We found that the depletion of either MRE11 or CtIP had a very clear impact on the etoposide sensitivity in WT cells but not in *ERCC6L2* deficient cells (**Fig. 4f, Supplementary Fig. 9e,f**). These results are similar to the epistatic relationship found with *NBN* (**Fig. 4d,e**). Investigating meta-analyses of DDR CRISPR screens, we found that *MRE11* and *RAD50* have never been observed hits in RPE1 cells (<https://sjlab.cruk.cam.ac.uk/app/ddrcs/>). This suggests that these knockouts either have a strong fitness effect on their own or there are technical problems with the library guide RNAs. False negatives are a known issue with many screening approaches, but our new individual experiments demonstrate that multiple components of the MRN complex and CtIP are indeed connected to the *ERCC6L2* phenotype.

Comment 3.3

Because the difference in RPA staining between WT versus *ERCC6L2* is very subtle, especially in S-phase, where most of the resection normally occurs (see Figure 3g), additional information supporting that *ERCC6L2* prevents hyperresection of the DSB would strengthen the manuscript. Does depletion of *ERCC6L2* enhance ATR signaling that is activated after resection of the break by monitoring RPA or Chk1 phosphorylation? Do *ERCC6L2* KO cells show increased levels of ssDNA at the break after ETP using BrdU detection (PMID: 34888531)? Can the author confirm that they don't see such changes after DNA damage-inducing DSB with blunt ends?

We agree with the reviewer that the RPA staining is very subtle in S phase after 1 h of etoposide treatment. To address this, we have now included cell cycle analysis in our experiment where we looked at RPA foci after letting the cells repair for 4 h following 30 minutes of etoposide treatment (**Fig. 3j**). In this late time point, we indeed saw a clear difference in both S and G2 phases comparing WT vs *ERCC6L2* KO cells.

We also monitored CHK1 phosphorylation as a marker for ATR activation and did not observe a clear difference between WT and *ERCC6L2* KO cells. We did not include this new data in the revised manuscript to maintain conciseness and clarity.

As suggested by this reviewer, to further demonstrate that *ERCC6L2* KO cells show higher resection upon staggered DSB induction, we have now monitored BrdU foci formation following etoposide treatment. We observed an increase in the induction of BrdU foci in *ERCC6L2* KO cells (**Supplementary Fig. 6k**). We additionally now performed a SMART assay to quantify resection track lengths, finding that *ERCC6L2* KO cells exhibit longer resection (**Fig. 3l** and **m**). These resection assays necessarily rely on conditions that introduce a very large number of DSBs, and so cannot be performed with a blunt DSB for direct comparison. Given that increased resection and all other phenotypes with etoposide mirror Cas12a-induced large deletions, we would expect that *ERCC6L2* loss would not lead to increased resection at a blunt Cas9 DSB. We have added mention of the experimental limitations in measuring resection after a Cas9 DSB to the text.

Comment 3.4

The interpretation of the results from the cell competition and clonogenic survival assays may be misleading, as they could reflect a severe growth defect rather than increased cell death. Can the authors demonstrate elevated cell death in *ERCC6L2*-deficient cells compared to WT using a cell death marker such as Annexin V or Caspase cleavage?

To address this, we used Annexin V staining for apoptosis. Using healthy donor and *ERCC6L2* deficient patient bone marrow stromal cells, we observed a significant increase in Annexin V positive staining in *ERCC6L2*^{-/-} + gMulti treated cells (**Fig. 3f**, **Supplementary Fig. 5h**). This suggests that *ERCC6L2* deficient cells undergo cell death upon induction of staggered DSBs rather than just a defect in proliferation.

Other comments:

Comment 3.5

There is a lack of statistical analysis across the manuscript. Moreover, if the analysis of the >1 kb deletions using sequencing data was obtained from a single replicate (as there is no error bar in these graphs), this should be indicated in the figure legend.

Statistical testing is now included for many of the analyses. Moreover, we have now performed biological replicates for all editing experiments in isogenic clones and patient-derived bone marrow stromal cells and obtained consistent results (**Fig. 2, Supplementary Fig. 2,3,4,7,8h**). This includes almost all long-read sequencing experiments. For assays with $n=1$, these bar graphs are clearly denoted with a single data point as well as mentioned in the figure legend.

Comment 3.6

In Figure 6d, the number of NBS1 foci/nucleus should be shown for each individual cell measured.

We thank the reviewer for this suggestion. NBS1 foci/nucleus formation from the experiment in the original manuscript is as follows:

We realize that the induction of NBS1 foci in this experiment is low. We speculate that this is likely because NBS1 forms visible foci only when it is extensively recruited.

To more accurately assess the recruitment of NBS1, we now used a proximity ligation assay (PLA) to measure the interaction of NBS1 with γ H2AX. The signal with PLA was much higher than measuring NBS1 foci. We therefore have replaced the NBS1 foci data with the PLA data (**Fig. 4i**). The conclusion remains unchanged, in that there was no difference in NBS1 recruitment to DSBs between WT and *ERCC6L2* KO cells. However, our ChIP-seq demonstrates that *ERCC6L2* knockout leads to increased MRE11 signal spreading surrounding Cas12a gMulti target sites (**Fig. 4k** and I). Taking these data together, this indicates that *ERCC6L2* does not affect MRN recruitment, but instead limits MRN spreading at DNA termini and prevents its function at more distal regions.

Comment 3.7

In addition, individual data points should also be indicated for Figures 3g, 3j, S5f, and S6g.

We have addressed this to show individual data points in the plots in new **Supplementary Fig. 6f, Fig. 4b, Supplementary Fig. 9a, Supplementary Fig. 6h**, respectively.

Comment 3.8

The legends at the top of the graph in Figure 3f and 3i should be made consistent (missing NBS^{-/-} in 3f)

We thank the reviewer for pointing this out. We now generated *NBN* KO clones and performed clonogenic assays to show phenotypes in a cleaner way than the previous competition assay, which used polyclonal population of heterogeneous *NBN* edited cells. In the revised manuscript, we replaced the old competition assay with the new data from *NBN* KO clones (**Fig. 4e**).

Comment 3.9

There is a missing data point in Figure 3d (*ERCC6L2* at 300nM ETP). Additionally, it would be beneficial to provide an example of clonogenic survival.

We apologize for the confusion. We didn't include 300nM ETP in *ERCC6L2* KO because there were zero colonies and the logarithmic y-axis means we cannot plot zeroes. We have now included representative images in new **Supplementary Fig. 6a** for the data quantified in old Fig. 3d (new **Fig. 3h**).

Comment 3.10

It is unclear how the *ERCC6L2* KO clones were validated by RT-qPCR. Is the author measuring mRNA expression (mRNA levels are often not affected by deletion events) or specific deletion events? This should be clearly indicated in the method section.

We apologize for any confusion. The *ERCC6L2* knockdown (KD) cells were validated by RT-qPCR (**Supplementary Fig. 1g**). We confirmed our *ERCC6L2* knockout (KO) isogenic clones via genotyping (**Supplementary Fig. 2a, 6b**). Due to the low endogenous expression of *ERCC6L2*, we found that we could not visualize *ERCC6L2* by Western blot using several antibodies. Instead, we have now performed genetic complementation experiments in the *ERCC6L2* KO cells with both WT and K154R *ERCC6L2* cDNA to determine whether the observed phenotypes are specific to *ERCC6L2* KO. These experiments used inducible FLAG-*ERCC6L2* lentiviral constructs (**Supplementary Fig. 3g, 6d**). Re-expression of WT *ERCC6L2* completely rescued the sequence loss phenotype (**Fig. 2f**) and etoposide hypersensitivity (**Fig. 3i**). Together, these new data

demonstrated that the phenotypes observed in the ERCC6L2 KO cells are specific to loss of function of ERCC6L2.

Comment 3.11

While Sanger sequencing was used to validate the NBS1 knockout clones, it remains unclear to what extent NBS1 is depleted in these clones. A Western blot analysis should be included to confirm protein loss, especially since, to my knowledge, reliable NBS1 antibodies are available, or by PCR

We thank this reviewer for this question and apologize for any confusion. For the competition assay experiment, we used a pooled population (not clones) of cells infected with a virus expressing a sgRNA targeting *NBN*. To more clearly validate screening results, we have now generated a *NBN* KO clone and assessed by Western blot the loss of NBS1 (**Supplementary Fig. 9c**). We used this clone to validate the epistasis with ERCC6L2 loss by a clonogenic assay (**Fig. 4e**). We have now replaced the old competition assay with these new data. Furthermore, we used this clone to measure RPA foci and found that the increased number of RPA foci observed in *ERCC6L2* KO cells depends on NBS1 (**Fig. 4g**).

Reviewer #4 (Remarks to the Author)

Reviewer #5 (Remarks to the Author):

Aird et al. investigate the differential requirements for repairing DSBs induced by different genome editing strategies, specifically comparing CRISPR-Cas9 and Cas12a, which produce blunt and staggered DSBs respectively. Using a clever dual-PAM reporter construct that enables delivery of both DSB types, combined with a tailored DDR gRNA library, they discovered that ERCC6L2 is preferentially required for protecting staggered DSB ends. The authors generated ERCC6L2 knockout and helicase-defective mutants and demonstrated through both short- and long-read sequencing that these cells accumulate large deletions and translocations. This protective role is independent of the nuclease generating staggered DSBs, as evidenced by similar phenotypes with Cas12a, TALEN-FokI, and dual Cas9 nickases. Through synthetic lethality screening with and without etoposide treatment, they found that while ERCC6L2-deficient cells are hypersensitive to etoposide, NBS1 (MRN complex member) depletion causes hypersensitivity in wild-type cells but does not further sensitize ERCC6L2-deficient cells, suggesting these factors operate in the same pathway. The authors demonstrate that ERCC6L2 limits ATM-dependent resection, as ATM inhibition reduces both resection and ERCC6L2-deficient cell sensitivity to staggered DSBs, while decreasing large deletion formation. Finally, biochemical analysis of purified ERCC6L2 protein reveals preferential binding to dsDNA and specific melting activity toward dsDNA substrates containing four-nucleotide overhangs, independent of overhang polarity (5' or 3').

The study is well-conducted and the data presented in this manuscript support the conclusion that ERCC6L2 prevents staggered DNA ends from forming large deletions and translocations. This is particularly relevant in etoposide-treated cells, which leave a 4-nucleotide overhang. My major experimental concerns involve the lack of complementation experiments with the clones and the use of an indirect measure of end resection, which is at the core of the proposed mechanism. Conceptually, I think the article lacks analysis of which pathway takes over in the ERCC6L2-deficient context, which has clinical relevance and could be easily addressed, and the fact that a more extensive discussion is needed of how the end-melting activity of ERCC6L2 would actually impede end-resection and the temporal relationship with ATM. Having said that, if the authors address these major concerns, I think readers of Nature Communications will enjoy this manuscript.

List of Major concerns:

Comment 5.1

1. In Figure 2, the authors generated two independent knockout clones but fail to provide complementation experiments in the analysis of deletion generation through sequencing. The authors should provide a complementation experiment with the full length protein.

Thank you for this suggestion. We have now performed genetic complementation experiments in the *ERCC6L2* KO cells with both the WT and K154R *ERCC6L2* cDNA to determine whether the observed phenotypes are specific to *ERCC6L2* KO. These experiments used inducible FLAG-*ERCC6L2* lentiviral constructs (**Supplementary Fig. 3g, 6d**). Re-expression of WT *ERCC6L2* completely rescued the sequence loss phenotype (**Fig. 2f**) and etoposide hypersensitivity (**Fig. 3i**). Together, these new data demonstrated that the phenotypes observed in the *ERCC6L2* KOs are specific to loss of function of *ERCC6L2*.

Comment 5.2

2. It is unclear which repair pathway takes over in *ERCC6L2*^{-/-} cells, which could have relevance for patients. Since POLQ appeared in their screen, do the authors think MMEJ is responsible for repairing breaks not protected by *ERCC6L2*? The authors should analyze the sequencing experiments for evidence of microhomology in the larger deletions and inhibit POLQ and test viability in the presence of staggered DSBs versus blunt DSBs. It would be interesting to test whether the translocations observed in BMSCs are MMEJ-dependent. This information is potentially important for patients with biallelic mutations in *ERCC6L2*.

We have now chemically inhibited factors involved in both canonical NHEJ and MMEJ. Inhibition of DNA-PKcs in a wild-type background expectedly led to increased sequence loss upon either Cas9 or Cas12a editing (**Supplementary Fig. 7a-c**). The absence of *ERCC6L2* further exacerbated these deletions (**Supplementary Fig. 7a-c**). However, *ERCC6L2* loss did not increase the chromosome arm loss caused by DNA-PKcs inhibition (**Supplementary Fig. 7d,e**). By contrast, inhibition of POLθ rescued kilobase-scale deletions in all conditions (**Supplementary Fig. 7a-c**). POLθ inhibition did not rescue megabase-scale deletions and chromosome arm loss in *ERCC6L2* KD cells (**Supplementary Fig. 7d,e**), demonstrating that multiple pathways are involved in repairing DNA damage in the absence of *ERCC6L2*.

Comment 5.3

3. The authors propose that *ERCC6L2* protects against resection mediated by CtIP. However, the only experimental setup in which they test this directly is RPA intensity after short and chronic etoposide treatment, which is indirect. The authors should provide another way to measure end-resection at breaks, for example by performing qPCR around the TRAC gRNA site to measure the extent of resection. If this proves difficult due

to Cas12a-induced ssDNA degradation (Chen et al., 2018) that could confound the results, the authors could use the AsiSI enzyme to induce breaks and measure resection at extensively validated AsiSI-induced breaks.

To address the referee's questions, we have now measured end resection with multiple assays:

- Using Single Molecule Analysis of Resection Tracks (SMART), *ERCC6L2* KO cells had an increased median resection length of 21.45 μm compared to 16.64 μm in wild-type cells following etoposide treatment (**Fig. 3l** and **m**).
- We measured increased BrdU focus formation following etoposide treatment in *ERCC6L2* KO cells relative to wildtype cells (**Supplementary Fig. 6k**).
- We used the suggested AsiSI qPCR readout and measured resection surrounding the cut sites. At 200 bp, we observed a significant increase in ssDNA content (**Supplementary Fig. 6l**). However, this increase was not observed at 950 or 1600 bp. Notably, AsiSI produces a short 2 bp overhang, while Cas12a, etoposide, and our staggered Cas9 nick experiments all produce longer overhangs. This could suggest that *ERCC6L2* has a minimal length requirement when protecting against long deletions. We do not have enough data to conclusively comment on this, but in the revised discussion we added a paragraph on this possibility.

Comment 5.4

4. While the authors propose that *ERCC6L2* prevents end-resection by melting DNA ends, how this melting prevents end-resection while favoring NHEJ is not entirely clear. How *ERCC6L2* is recruited to breaks and what the temporal relationship is between *ERCC6L2* end melting and ATM-dependent resection remains unclear. Does *ERCC6L2* arrive before MRN, perform the melting, and thereby impede ATM activation and subsequent end-resection? The authors should at least discuss this in the discussion.

We thank this reviewer for this suggestion. We have now included a clearer model in the discussion. We propose that, during NHEJ, *ERCC6L2* protects the staggered ends by melting them and promoting efficient end-pairing and ligation. This is in line with several studies that proposed some degree of DNA end unwinding stabilizes synapsis and promotes DNA-PKcs activity during NHEJ (Pawelczak et al. 2005) (DeFazio LG et al. 2002) (Jovanovic M et al. 2006). In the absence of *ERCC6L2* melting activity, NHEJ would thus be less efficient, which would provide greater opportunities for the MRN complex to initiate resection through an ATM mediated process. This is also consistent with *ERCC6L2* being recruited to damage sites immediately after KU and XLF, and prior to NBS1 (Liu X, et al. 2020).

Comment 5.5

5. Throughout the article, consideration of the cell cycle is missing. Do ERCC6L2-deficient cells show an impaired cell cycle that changes the repair pathway? Is ERCC6L2 activity regulated during the cell cycle? Is this melting activity present only during S-phase, or do the authors think it might also work in G1? This should at least be taken into consideration during the discussion.

We now measured that both the cell cycle status (**Supplementary Fig. 2b**) and growth rate (**Supplementary Fig. 2b**) of *ERCC6L2* KO cells are unaffected compared to wild-type cells (*TP53* deficient background). We furthermore measured RPA foci data in different cell cycle phases when we let the cells repair for 4 h following 30 min of etoposide treatment. We observed a clear difference in both S and G2 phases comparing WT vs *ERCC6L2* KO cells, suggesting that increased resection observed in *ERCC6L2* deficient cells is not due to a difference in cell cycle distribution (**Fig. 3j**).

The lack of increased RPA foci during G1 is likely due to the regulation of CtIP expression and/or phosphorylation status rather than a lack of *ERCC6L2* activity in G1. An RPA focus is only visible when resection is long enough to accommodate sufficient RPA complexes (Huertas P et al. 2015). Indeed, *ERCC6L2* KO also shows an increase in small deletions after Cas12 induction of DSBs (**Fig. 2a,c, Supplementary Fig. 2g,k**), which could be due to the activity of the MRN complex at DSBs during G1 when they do not undergo extensive resection. Discussion of this point has been included in the manuscript.

Minor comments:

Comment 5.6

1. In figure 1f would it be worth to include also the 0 bp events, that account for both perfect NHEJ or unedited events? Having the 0bp events will make it easier for the reader to read the graphs normalized by the cutting efficiency.

Below we show indel plots that include 0 bp events at the *TRAC* locus in both K-562 and RPE1 cells. In our opinion, the large number of bars/colors when including 0 bp events makes the plots more difficult to read, especially in the case when replicates have different editing efficiencies. We note that the profiles of indel outcomes are consistent regardless of editing efficiency (example in comparing **Supplementary Fig. 3d** and **3e**), which highlights the consistency of the indel phenotype. We therefore continued to display the modified indel outcomes and editing efficiency graphs separately in the revised manuscript, but could combine them if the referee and editor feel strongly about this.

Comment 5.7

2. In Figure 3, the authors found NBS1, one subunit of the MRN complex, but not MRE11 or RAD50 or CtIP. While CtIP is commonly essential in DepMap and MRE11 is probably essential in RPE1, RAD50 should not be. Why were they not discovered in the screen? Can the authors perform another validation curve as in panel e, inhibiting both ERCC6L2 and at least one other MRN complex subunit to verify rescue of etoposide sensitivity?

To address these questions, we have now used siRNA knockdown of MRE11 and CtIP. We depleted either *MRE11* or *RBBP8* (CtIP) via siRNA knockdown in *ERCC6L2* KO or WT cells. We found that the depletion of either MRE11 or CtIP had a very clear impact on the etoposide sensitivity in WT cells but not in *ERCC6L2* deficient cells (**Fig. 4f**, **Supplementary Fig. 9e,f**). These results are similar to the epistatic relationship found with *NBN* (**Fig. 4e**). Investigating meta-analyses of DDR CRISPR screens, we found that *MRE11* and *RAD50* are not observed hits in RPE-1 cells (<https://sjlab.cruk.cam.ac.uk/app/ddrcs/>). This suggests that these knockouts either have a strong fitness effect on their own or there are technical problems with the library guide RNAs. False negatives are a known issue with many screening approaches, but our new individual experiments demonstrate that multiple components of the MRN complex and CtIP are indeed involved in the *ERCC6L2* phenotype.

Comment 5.8

3. In fig 3a the author showed that multiple DSBs induce growth defects in *ERCC6L2*^{-/-}, contrary to a single DSB induced in Fig. 1d. The author suggested that the difference in viability outcome is due to the fact that multiple DSBs induce large genomic aberrations, while a single DSB only induce large deletions. But in figure 2j the author measured translocation in *ERCC6L2* KO with a single DSB. The author should sustain this statement by analyzing chromosomal aberrations on a metaphase spreads in both

condition (1 gRNA vs gMulti), (or any other experimental set up they like) to test this hypothesis.

We have now characterized gMulti in more depth. First, we microscopically inspected cells for micronuclei which arise from unrepaired DNA damage that leads to mis-segregation of chromosomes or acentric chromosomal fragments in mitosis (Fenech M. 2000). Editing with gMulti significantly increased the number of *ERCC6L2*^{-/-} cells harboring micronuclei ($64.8 \pm 11.6\%$) compared to wild-type cells ($33.3 \pm 10.8\%$) (**Fig. 3c,d**).

Furthermore, we edited patient BMSCs with gMulti. We observed significantly fewer surviving colonies in *ERCC6L2* patient cells compared to the healthy donor controls (**Fig. 3e, Supplementary Fig. 5f,g**). We also quantified significantly more cells that stained annexin V positive, an indicator of apoptosis (**Fig. 3f, Supplementary Fig. 5g,h**). Together, these data indicate that the difference in viability in gMulti-edited *ERCC6L2* KO versus wild-type cells is indeed due to large genomic aberrations and active cell death rather than just a competitive outgrowth.

Comment 5.9

4. Fig3.d some error bars are missing, and one is black other than grey.

We apologize for the confusion. Error bars are not visible in WT at low doses in this figure (now **Fig. 3h**) due to small standard deviations and the logarithmic scale. On a linear scale, error bars are more visible in WT at low doses. For clarity, we continue to show the logarithmic scale graph in the revised manuscript.

Comment 5.10

5. Fig. Ex2a the TIDE analysis on the left panel is barely visible, increase the size or remake the graph.

We thank this reviewer for pointing this out. We remade the TIDE genotype graphs to be more readable (**Supplementary Fig. 2a, 6b, 6m, 8a-c, 9d**).

Comment 5.11

6. In Figure 3d, the authors measure growth of ERCC6L2 knockouts with or without etoposide in p53-positive or p53-negative cells. Do the knockouts grow similarly to wild-type cells in p53+ and p53- contexts? It would be helpful to provide growth curves showing the difference in Extended Figure 5, along with cell cycle analysis to determine whether ERCC6L2 knockout impairs cell cycle progression and therefore repair pathway utilization.

We have now performed the suggested cell cycle analysis and growth curves in *TP53* positive and negative cells. We found that in *TP53* proficient cells, the loss of ERCC6L2 caused a lower growth rate. However, this was not observed in *TP53* deficient cells (Supplementary Fig. 2c and reproduced below).

Our manuscript almost exclusively uses the *TP53* deficient cells that exhibit no change in growth rate. We only used the *TP53* proficient background to validate that etoposide hypersensitivity and increased RPA recruitment in *ERCC6L2* deficient background were independent of *TP53* status. To exclude the possibility that altered growth rate affects RPA recruitment, we analysed RPA foci across the different phases of the cell cycle when we let the cells repair for 4 h following 30 min of etoposide treatment (Fig. 3j), as previously mentioned in comment 5.5. We observed a clear difference in both S and G2 phases comparing WT vs *ERCC6L2* KO cells, suggesting that increased resection observed in *ERCC6L2* deficient cells is not due to a difference in cell cycle distribution (Fig. 3j).

Comment 5.12

7. In Figure 6d, the authors used a short but very high dose of etoposide to monitor NBS1 foci formation. On average, they observed 2-3 foci per nucleus. While I agree with the interpretation that there seems to be no difference in repair of those breaks with or without ERCC6L2, the large error bars and the fact that breaks are not fully repaired suggest these conditions are not ideal for testing this. How many breaks did the authors expect with this treatment? Can the authors provide γ H2AX data under the same conditions? Can they use gMulti to induce multiple breaks that should give a higher number of NBS1

foci? Or how many phospho-RPA foci did they observe at 3 hours? While not identical conditions, this would give readers an idea about the expected number of DSBs.

We thank the reviewer for pointing this out. We observed a strong γ H2AX signal despite the limited number of NBS1 foci:

We speculate that this is likely because NBS1 forms visible foci only when it is extensively recruited, whereas γ H2AX spreading occurs at all DSBs. To better assess the recruitment of NBS1, we have now used a proximity ligation assay (PLA) to the interaction of NBS1 with γ H2AX. We found that the PLA signal was much stronger than NBS1 focus formation alone, and in the revised manuscript we replaced the NBS1 focus data with PLA data (**Fig. 4i**), and therefore we didn't include neither the γ H2AX shown above. The conclusion from the PLA data remained unchanged in that there was no difference in NBS1 recruitment to DSBs between WT and *ERCC6L2* KO cells. However, ChIP-seq for MRE11 demonstrated an increased MRE11 signal spreading surrounding Cas12a gMulti target sites in *ERCC6L2* KO cells (**Fig. 4k** and **I**). This suggests that *ERCC6L2* does not impact MRN recruitment, but instead limits MRN spreading at DNA termini and prevents its function at more distal regions.

Response to reviewers' comments on the revised manuscript

Reviewer #1 (Remarks to the Author):

The authors have addressed the reviewers' concerns regarding reproducibility, statistical testing, the limited role of ERCC6L2 in NHEJ, and have added an analysis of the role of ERCC6L2 in the absence of NHEJ. The authors have also corrected other minor errors. The reviewer believes that this manuscript is now ready for publication in Nature Communications.

We thank this reviewer for their positive reception of the revised manuscript.

Reviewer #2 (Remarks to the Author):

Reviewer #3 (Remarks to the Author):

The authors have addressed all of my comments thoroughly. I strongly support the publication of this manuscript in Nature Communications.

We thank this reviewer for their positive reception of the revised manuscript.

Minor comments:

Why is there such an extensive BrdU signal in the SMART assay for untreated samples, which should not exhibit double-strand breaks leading to DNA end resection?

We thank the reviewer for raising this point which we are happy to have the opportunity to clarify. Untreated control samples were included in these experiments; however, we did not detect BrdU signal which we could measure via the SMART assay, as the reviewer notes. For clarity, we presented conditions in which ssDNA is induced and quantifiable (i.e. upon etoposide treatment).

The data shown compare WT and *ERCC6L2*^{-/-} cells upon etoposide treatment as well as etoposide treatment followed by a 4h recovery, allowing us to visualise ssDNA generated during the processing of etoposide-induced TOP2ccs. The increase in BrdU-positive ssDNA therefore reflects post-treatment processing of etoposide-induced TOP2ccs, rather than baseline ssDNA present in untreated cells (**Fig. 3m**).

Reviewer #4 (Remarks to the Author):

Reviewer #5 (Remarks to the Author):

The authors addressed all my comments with solid new experiments and an improved discussion. The paper is now suitable for publication.

We thank this reviewer for their positive reception of the revised manuscript.